



# Responses of Arctic Black Carbon and Surface Temperature to Multi-Region Emission Reductions: an HTAP2 Ensemble Modeling Study

**Na Zhao[1], Xinyi Dong[2], Joshua S. Fu[3,4*], Marianne Tronstad Lund[5], Kengo Sudo[6], Daven Henze[7], Tom Kucsera[8], Yun Fat Lam[9], Mian Chin[10], Simone Tilmes[11], Kan Huang[1]***

[1] Shanghai Key Laboratory of Atmospheric Particle Pollution and Prevention (LAP3), Department of Environmental Science and Engineering, Fudan University, Shanghai, China

[2] School of Atmospheric Science, Nanjing University, Nanjing, China

[3] Department of Civil and Environmental Engineering, The University of Tennessee, Knoxville, Tennessee, USA

[4] Computational Earth Science Group, Computational Sciences and Engineering Division, Oak Ridge National Laboratory, Oak Ridge, Tennessee, USA

[5] CICERO Center for International Climate and Environmental Research, Oslo, Norway

[6] Nagoya University, Furo-cho, Chigusa-ku, Nagoya, Japan

[7] Department of Mechanical Engineering, University of Colorado, Boulder, CO, USA

[8] Universities Space Research Association, Greenbelt, MD, USA

[9] Department of Geography, The University of Hong Kong, HKSAR, China

[10] Earth Sciences Division, NASA Goddard Space Flight Center, Greenbelt, MD, USA

[11] Atmospheric Chemistry Observations and Modeling Laboratory, National Center for Atmospheric Research, Boulder, Colorado, USA

Correspondence: jsfu@utk.edu; huangkan@fudan.edu.cn

## ABSTRACT

Black carbon (BC) emissions play an important role in regional climate change of the Arctic. It is necessary to pay attention to the impact of long-range transport from regions outside the Arctic as BC emissions from local sources in the Arctic were relatively small. The Task Force Hemispheric Transport of Air Pollution Phase2 (HTAP2) set up a series of simulation scenarios to investigate the response of BC in a given region to different source regions. This study investigated the responses of Arctic BC concentrations and surface temperature to 20% anthropogenic emission reductions from six regions in 2010 within the framework of HTAP2 based on ensemble modeling results. It was found that the emissions from East Asia (EAS) had most (18.1%–51.4%) significant impact on the Arctic



while the monthly contributions from Europe, Middle East, North America, Russia Belarus Ukraine,
and South Asia were 20.1%–49.9%, 0.02%–0.9%, 8.3%–19.3%, 5.4%–18.1%, and 3.1%–7.7%,
respectively. The responses of the vertical profiles of the Arctic BC to the six regions were found to be
different due to multiple transport pathways. The response of the Arctic BC to emission reductions of
six source regions became less significant with the increase of the latitude. The benefit of BC emission
reductions in terms of slowing down surface warming in the Arctic was evaluated by using Absolute
Regional Temperature-change Potential (ARTP). Compared to the response of global temperature to
BC emission reductions, the response of Arctic temperature was substantially more sensitive,
highlighting the need for curbing global BC emissions.

## 1. Introduction

Black carbon (BC) is one of the short-lived climate forcers (SLCFs, AMAP, 2015) and was regarded
as the second largest contributor to global warming, only inferior to carbon dioxide (Bond et al. 2013).
BC over the Arctic can perturb the radiation balance in a number of ways. Direct aerosol forcing
occurred through absorption or scattering of solar (shortwave) radiation. BC is the most efficient
atmospheric particulate species at absorbing visible light (Bond et al. 2013), the added atmospheric
heating will subsequently increase the downward longwave radiation to the surface and warm the
surface (AMAP, 2011). Radiative forcing by BC can also result from aerosol-cloud interactions that
affected cloud microphysical properties, albedo, extent, lifetime, and longwave emissivity (Twomey
1977; Garrett and Zhao 2006). BC has an additional forcing mechanism after depositing onto snow
and ice surfaces (Clarke and Noone, 1985). The surface albedo of snow and ice could be reduced and
further enhanced the absorption of solar radiation at the surface. In the Arctic, surface temperature
responses were strongly linked to surface radiative forcing as the stable atmosphere of the region
prevented rapid heat exchange with the upper troposphere (Hansen and Nazarenko, 2004).
The Arctic has been warming twice as rapidly as the world in the past fifty years, and has
experienced significant changes in its ice and snow covers as well as permafrost (AMAP, 2017).
Reductions of carbon dioxide emissions are the backbone of any meaningful effort to mitigate climate
forcing. But even if significant reductions of carbon dioxide are made, slow down of the temperature
rise in the Arctic and the sea level rise caused by the melting of glaciers may not be achieved in time.



Hence, the goal of slowing down the deterioration of the Arctic may best be achieved by also targeting
at shorter-lived climate forcing agents, especially those that could impose appreciable surface forcing
and trigger regional-scale climate feedbacks pertaining to the melting of sea ice and snow. Modelling
studies by UNEP/WMO (2011) and Stohl et al. (2015) suggested that the climate response of SLCFs
mitigation was strongest in the Arctic region. AMAP (2011 and 2015) as well as Sand et al. (2016)
demonstrated that per unit of emission reductions of SLCFs in the Northern areas had the largest
temperature response on the Arctic, with the Nordic countries (Denmark, Finland, Iceland, Norway,
and Sweden) and Russia having the largest impact compared to other Arctic countries such as the
United States and Canada.

The few studies that investigated specific regional aerosol forcing (Shindell and Faluvegi, 2009;

Shindell et al., 2012; Teng et al., 2012) typically used a single climate model at a time to investigate
the climate response to idealized, historical, or projected forcing. However, different models varied
considerably in the representation of aerosols and radiative properties, resulting in large uncertainties
in simulating the aerosol radiative forcing (Myhre et al., 2013b; Shindell et al., 2013). When
investigating the climate response to regional emissions, such uncertainties were likely to be
confounded even further by the variability between models in regional climate and circulation patterns
and variation in the global and regional climate sensitivity (the amount of simulated warming per unit
radiative forcing). Hence, the Task Force Hemispheric Transport of Air pollution Phase2 (HTAP2,
http://www.htap.org/) incorporating multiple global models can avoid the great uncertainty of single
model to a certain degree, with the aim to improve model estimates of the impacts of intercontinental
transport of air pollutants on climate, ecosystems, and human health (Galmarini et al., 2017). To date,
the HTAP2 results have been explored from a variety of scientific and policy-relevant perspectives.
For instance, by comparing against observations, sulfur and nitrogen depositions during HTAP2 had
been significantly improved compared to HTAP1. From 2001 to 2010, the global nitrogen deposition
increased 7% while the global sulfur deposition decreased 3% (Tan et al., 2018a). The significant
impacts of hemispheric transport on the deposition were specifically focused and the deposition over
the coastal regions was more sensitive to hemispheric transport than the non-coastal continental
regions (Tan et al., 2018b). Jonson et al. (2018) assessed the contributions from different world regions
to European ozone levels and contributions from the non-European regions were mostly from North

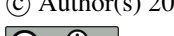



America and eastern Asia, larger than those from European emissions. Hogrefe et al. (2018) found that
the simulated ozone over the continental US varied very differently by digesting boundary conditions
from four hemispheric or global models. The impact of emission changes from six major source
regions on global aerosol direct radiative forcing was estimated (Stjern et al., 2016). In the local source
regions, the radiative forcing associated with $SO_4^{2-}$ was strengthened (25%) while that from BC was
weakened (37%) due to a 20% emission reduction. Liang et al. (2018) estimated global air-pollution-
related premature mortality from exposure to $PM_{2.5}$ and ozone and the interregional transport lead to
more deaths through changes in $PM_{2.5}$ than in $O_3$. However, the source region contributions to Arctic
BC and the spread among multi-model results have been rarely explored from the perspective of
HTAP2 initiative.
This study aims to investigate the responses of Arctic BC concentrations and surface temperature to
20% anthropogenic emission reductions from different regions in the Northern Hemisphere. A
comparison of six global modeling works within the framework of HTAP2 experiments for the Arctic
region in 2010 was presented. The ensemble modeling results were used to apportion the contribution
from different source regions to the near-surface and vertical black carbon in the Arctic. In addition,
the Arctic surface temperature responses to the emission reductions were estimated.

## 2. Methodology

### 2.1 HTAP2 experiments

HTAP2 developed a harmonized emissions database covering all countries and the major sectors for
global and regional modeling from 2008 to 2010. The emissions database was obtained from the
nationally reported emissions (e.g., National Emission Inventory for the United States), the regional
scientific inventories (e.g., Model Inter-Comparison Study for Asia, MICS–Asia III), and the
Emissions Database for Global Atmospheric Research data (EDGARv4.3, emissions for South
America, Africa, Russia and Oceania). Biomass burning emissions were not prescribed in HTAP2. It
was recommended that modeling groups used the Global Fire Emissions Database (GFED4,
http://globalfiredata.org/) with a temporal resolution of daily or 3–hour intervals. The detailed
information of different regional inventories can be found in (Janssens–Maenhout et al., 2015).



Emission perturbations were conducted in sensitivity simulations to investigate the response of
various air pollutants in a given region to different source regions. In this study, the Arctic region was
the targeted receptor region of interest. Six source regions in HTAP2 experiments, namely, East Asia
(EAS), Europe (EUR), Middle East (MDE), North America (NAM), Russia–Belarus–Ukraine (RBU),
and South Asia (SAS) were selected to demonstrate their influences on the BC concentrations over the
Arctic region (Figure 1a). Two emission scenarios were designed for the HTAP2 simulation to explore
the source/receptor relationships, i.e. the base scenario (BASE) with no emission reduction, and the
control scenario (EASALL, EURALL, MDEALL, NAMALL, RBUALL, and SASALL) with 20%
reduction of all anthropogenic emissions in six regions, respectively.

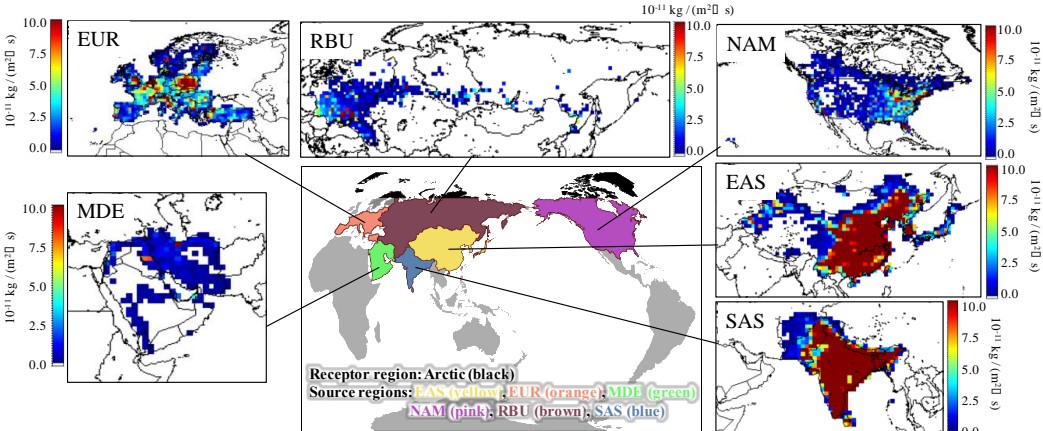


**Figure 1. (a)** The sketch map of receptor and source regions. **(b)–(g)** Spatial distributions of 20% reduction of annual
BC emission in the six source regions in 2010. EAS: East Asia; EUR: Europe; MDE: Middle East; NAM: North
America; RBU: Russia–Belarus–Ukraine; SAS: South Asia. The unit legends from (b) to (g) are the same of $10^{-11}$
kg (m² s)⁻¹.

### 2.1.1 Anthropogenic emission reductions of BC in HTAP2

Anthropogenic BC emission sectors included power plants, industries, transportation, shipping,
aviation, agriculture, and residential sectors. The emission inventory had a monthly temporal
resolution and a spatial resolution of 0.1° × 0.1°. The total anthropogenic emissions and 20% emission
reductions of BC in six source regions of HTAP2 in 2010 are presented in Table 1. The higher BC

On

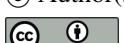



emission reductions were found in EAS and SAS with the values of 355.6 and 232.5 Gg yr$^{-1}$,
respectively, while were much lower in MDE and RBU with the values of 5.3 and 18.6 Gg yr$^{-1}$,
respectively. The BC emission reductions in EAS, EUR, and RBU indicated significant monthly
variations with higher values from November to March, while the monthly variations were not obvious
in MDE, NAM, and SAS.

**Table 1.** 20% emission reductions and total anthropogenic emissions of BC in different regions of HTAP2 in 2010.
(Unit: Gg yr$^{-1}$).

| Regions | Total anthropogenic emissions | 20% Emission Reductions | | | | | | | | | | | |
|---|---|---|---|---|---|---|---|---|---|---|---|---|---|
| | | Feb | Mar | Apr | May | Jun | Jul | Aug | Sep | Oct | Nov | Dec | All |
| EAS[a] | 1778.1 | 35.6 | 33.0 | 24.1 | 23.9 | 24.0 | 24.0 | 23.6 | 23.5 | 25.0 | 31.4 | 41.0 | 355.6 |
| EUR[b] | 326.3 | 6.4 | 7.2 | 6.5 | 5.3 | 4.9 | 4.0 | 3.7 | 4.4 | 5.2 | 5.3 | 5.7 | 65.3 |
| MDE[c] | 26.7 | 0.4 | 0.5 | 0.5 | 0.5 | 0.4 | 0.4 | 0.4 | 0.4 | 0.5 | 0.5 | 0.5 | 5.3 |
| NAM[d] | 310.8 | 5.1 | 5.3 | 5.1 | 5.1 | 5.2 | 5.3 | 5.3 | 5.1 | 5.1 | 5.1 | 5.2 | 62.2 |
| RBU[e] | 93.0 | 1.9 | 1.9 | 1.7 | 1.4 | 1.3 | 1.0 | 1.1 | 1.2 | 1.6 | 1.7 | 1.8 | 18.6 |
| SAS[f] | 1162.7 | 19.1 | 19.7 | 18.9 | 19.2 | 18.9 | 19.2 | 19.2 | 19.0 | 19.3 | 19.2 | 20.4 | 232.5 |
| All | 3697.6 | 68.5 | 67.6 | 56.8 | 55.4 | 54.7 | 53.9 | 53.3 | 53.6 | 56.7 | 63.2 | 74.6 | 739.5 |

[a] East Asia. [b] Europe. [c] Middle East. [d] North America. [e] Russia–Belarus–Ukraine. [f] South Asia.

Figure 1b–g illustrates the spatial distribution of the 20% reductions of annual BC emissions in six
source regions in 2010. It can be found that the most intense reductions of BC emissions in EAS and
SAS were concentrated in East China and India, respectively, which were mainly attributed to
emissions from residential sectors, followed by transportation and industries. The BC emission
reductions of EUR were widely distributed with high values in central Europe, with residential and
transportation sectors accounting for the largest proportions. The reductions near the Arctic circle could
be found in the north of EUR, NAM, and RBU. For MDE, most BC was emitted from Iran, which
located in the northeast of this region. Overall, the spatial pattern of BC emission reductions in six
regions was closely related to the distribution of human populations.

**2.1.2 Model description**
In this study, six global models (CAMchem, CHASER_re1, CHASER_t106, GEOS-Chem,
GOCART–v5, and OsloCTM3–v2) in BC experiment were incorporated to simulate the responses of





BC concentrations in the Arctic to 20% BC emission reductions of the EAS, EUR, MDE, NAM, RBU,
and SAS, respectively. The brief information of model configurations is listed in Table 2. As required
by HTAP2, all simulations should include a spin-up time of 6 months prior to the period of interest.
The outputs from all models are available upon request from http://aerocom.met.no. The time
resolution of the outputs used in this study is monthly for all models, although models were run at a
finer resolution (e.g., daily or hourly). The model outputs for air pollutants were originally provided
in the unit of mass mixing ratio (MMR, kg kg$^{-1}$). To facilitate comparison between model and
observation and further data analysis, we converted the original units into ng m$^{-3}$ based on the ideal
gas law (Aamaas et al., 2017).

**Table 2.** Configurations of models used in this study

| Models | Meteorological field | Horizontal resolution | Vertical layers | Convection | Reference |
|---|---|---|---|---|---|
| CAMchem | GEOS5 v5.2 | 1.9° × 2.5° | 56 | Zhang–McFarlane approach for deep convection | Lamarque et al., 2012; Tilmes et al., 2016 |
| CHASER_re1 | ERA-Interim and HadISST | 2.8° × 2.8° | 32 | CCSR/NIES AGCM for advection, convection, and other subgrid–scale mixing | Sudo et al., 2002; Takashi et al., 2018 |
| CHASER_t106 | ERA-Interim and HadISST | 1.1° × 1.1° | 32 | CCSR/NIES AGCM for advection, convection, and other subgrid–scale mixing | Sudo et al., 2002; Takashi et al., 2018 |
| GEOS-Chem | GEOS–5 (MERRA) | 2.0° × 2.5° | 47 | Convective transport is computed from the convective mass fluxes in the meteorological archive | Henze et al., 2007 |
| GOCART–v5 | MERRA | 1.3° × 1.0° | 72 | MERRA for moist convection, Arakawa–Schubert (RAS) algorithm for GCTM | Chin et al., 2000 |
| OsloCTM3–v2 | ECMWF–IFS | 2.8° × 2.8° | 60 | Tiedke mass flux scheme for deep convection | Søvde et al., 2012; Lund et al., 2018 |


## 2.2 Calculation of the temperature response to BC emissions reduction

The climate effects of air pollutants have been the focus of climate change research since the last
century (IPCC, 1990; IPCC, 2001). In the last few years, the metrics for estimating this kind of effect
have been constantly improving (Shindell et al., 2012; Bond et al., 2013; Smith and Mizrahi, 2013;



Stohl et al., 2015). The Intergovernmental Panel on Climate Change (IPCC) used the Global Warming
Potential (GWP) as a method for comparing the potential climate impact of emissions of different
greenhouse gases (IPCC, 1990). GWP is the time-integrated radiative forcing due to a pulse emission
of a given species, over some given time horizon (commonly 20, 100, or 500 years) relative to a pulse
emission of carbon dioxide. GWP does not purport to represent the impact of air pollutant emissions
on temperature. Although a short-lived climate pollutant (SLCP) could have the same GWP as a long-
lived climate pollutant, identical (in mass terms) pulse emissions could cause a different temperature
change at a given time, because long-lived climate pollutants accumulate in the climate system while
short-lived climate pollutants can be broken down by various processes. Consequently, warming
caused by long-lived climate pollutants is determined by total cumulative emissions to date, while the
warming due to short-lived climate pollutants is determined more by the current rate of emissions in
any given decade and depends much less on historical emissions. This means the importance of SLCP
emissions is often overstated based on GWP. Shine et al. (2005) proposed the Global Temperature
Change Potential (GTP) as a replacement for GWP to represent the global-mean surface temperature
change for both a pulse emission ($GTP_P$) and a sustained change in emissions ($GTP_S$) of a given air
pollutant. The distinction between $GTP_P$ and $GTP_S$ avoids the overestimation of GWP for the short-
lived climate pollutants. Even for a uniform forcing, there will be differences of spatial patterns in the
temperature response. Regional Temperature-change Potential (RTP) (Shindell and Faluvegi, 2010)
was applied to analyze the temperature response on the regional scale, since both GWP and GTP
focused on the global scale. The GWP, GTP, and RTP were normalized to the corresponding effect of
$CO_2$ as the Absolute Global Warming Potential (AGWP), Absolute Global Temperature Change
Potential (AGTP), and Absolute Regional Temperature-change Potential (ARTP), respectively. AGWP
represented the absolute forms of radiative forcing. AGTP and ARTP represented the absolute forms
of temperature perturbation.
ARTPs is more suitable for this study to calculate the temperature response, considering that the
research object is BC with short lifetime and focus on regional impact of the BC emission reductions
on temperature changes in the Arctic. For SLCFs with atmospheric lifetimes much shorter than both
the time horizon of the ARTP and the response time of the climate system, the general expression for
the ARTP following a pulse emission of BC ($E$) in region $r$ which leads to a response in latitude band





*m* is as follows (Fuglestvedt et al., 2010; Collins et al., 2013; Aamaas et al., 2017):
$\mathrm{ARTP}_{r,m,s}(H) = \sum_l \frac{F_{l,r,s}}{E_{r,s}} \times \mathrm{RCS}_{l,m} \times R_T(H)$                    (1)
$F_{l,r,s}$ (in W m$^{-2}$) is the radiative forcing in latitude band *l* due to emission in region *r* in season *s* as a
function after the pulse emission $E_{r,s}$ (in Tg). Even though our estimates are based on seasonal
emissions, the temperature responses calculated are annual means. Shindell and Faluvegi (2009)
analyzed BC climate effect in four different latitudes: southern mid-high latitudes (90°S–28°S),  tropics
(28°S–28°N), northern mid-latitudes (28°N–60°N), and the Arctic (60°N–90°N), which gives a better
estimate of the global temperature response as it accounts for varying efficacies with latitude. The
$RCS_{l,m}$ is a matrix of regional response coefficients based on the RTP concept (unitless; Collins et al.,
2013). As these response coefficients are normalized here, they contain no information on climate
sensitivity, only the relative regional responses in the different latitude bands. The global climate
sensitivity is included in the impulse response function R$_T$, which is a temporal temperature response
to an instantaneous unit pulse of RF (in K m$^2$ W$^{-1}$). This paper refers to the ARTP values in Aamaas et
al. (2017). Aamaas et al. (2017)  applies two refinements of the forcing-response coefficients for
radiative forcing occurring in the Arctic: one for the aerosol effects in the atmosphere (Shindell and
Faluvegi, 2010; Lund et al., 2014) and another for the effects due to BC on snow (Flanner, 2013). The
ARTP in this study estimated of the direct effect in the Arctic included both the direct radiative forcing
from outside the Arctic and within the Arctic, while the ARTP of the semi-direct effect in the Arctic
was due to the semi-direct radiative forcing from outside the Arctic. The contribution by radiative
forcing within the Arctic to Arctic temperature changes considered the vertical profile of BC
concentrations as both $F_{Arctic,r,s}$ and $RCS_{Arctic,Arctic}$ have a dependence on the height of the BC (Lund et
al., 2014; Lund et al., 2017). The total response in the Arctic was the sum of the contributions from
BC forcing outside of the Arctic and inside of the Arctic.
Regional temperature responses at time *t* of an emission *E*(t) can be calculated with these ARTP
values by a convolution (Aamaas et al., 2016). The temperature response is as follows:
$\Delta T_{r,m,s,t}(t) = \int_0^t E_{r,s,t}(t') \times \mathrm{ARTP}_{r,m,s,t}(t-t')dt'$                    (2)
$\Delta T_{r,m,s,t}$ refers to the decrease of the Arctic or global surface temperature after 20, 100, or 500 years
to 20% BC emission reductions of six regions (namely EAS, EUR, MDE, NAM, RBU, and SAS) in



the framework of HTAP2 either during summer or winter.

**3. Results and Discussion**

**3.1 Model evaluation**

To evaluate the model performance from all six models, the monthly simulated surface BC
concentrations of the BASE case were compared with the observations at four monitoring sites in the
Arctic Circle in 2010. The locations of the four sites, including Alert (82.5°N, 62.3°W) in Canada,
Barrow (71.3°N, 156.6°W) in Alaska, Tiksi (71.59°N, 128.92°E) in Russia, and Zeppelin (78.9°N,
11.9°E) in Norway, are plotted in Figure S1 in the Supporting Information.
Metrics (Text S1) including correlative coefficient (COR), normalized mean bias (NMB),
normalized mean error (NME), mean bias (MB), and mean absolute error (MAE) were selected for
evaluating the model performance in this study (U.S. EPA, 2007) In addition to the evaluation for each
single model, the multi-model ensemble mean (calculated as the average of all participating models)
was also evaluated. The statistical results are listed in Table 3 and Table S1. A comparison between the
temporal variations of simulated and observed BC concentrations is shown in Figure S2.
The correlations of the simulated BC concentrations among different models were moderate to high
with CORs ranging from 0.33 to 0.98 (Table S1), suggesting the temporal variations of different
models were relatively consistent. Overall, CAMchem, GEOS-Chem, GOCART–v5, and OsloCTM3–
v2 underestimated the near-surface BC (Figure S2), which may be attributed to an underestimation of
BC emissions, e.g., gas flaring (Huang et al., 2014, 2015; Stohl et al., 2013) and shipping emissions
(Marelle et al., 2016). However, the simulated BC surface concentrations from CHASER_re1 and
CHASER_t106 were higher than those of the other four models and observations (Figure S2), which
were mainly due to their slow BC aging-rate in remote/polar regions (Sudo et al., 2015).
Table 3 shows the model performances at the four Arctic sites. Relatively good agreement between
the observation and models was found at Zeppelin, with CORs, NME, MB, and MAE of 0.59–0.83,
38.5%–142.6%, –13.5–15.0 ng m$^{-3}$, and 5.4–15.0 ng m$^{-3}$, respectively. On the contrary, the simulated
BC concentrations didn't agree so well with observations at the other three sites with even negative
COR values in some models (e.g., CAMchem, CHASER_re1, and CHASER_t106), which may be





explained by the uncertainties in emission inventory, the bias in the meteorological simulations, and
chemical mechanisms (Miao et al., 2017; Zhang et al., 2019). All models, except OsloCTM3,
overestimated the BC concentrations in Barrow in July (Figure S2), mainly due to the large
contributions of biomass burning from Siberia in the simulations caused by overestimations of
emissions and/or too little removal during transport (Sobhani et al., 2018). Although the single model
didn't reproduce the BC concentrations in the Arctic well, the consistency of the model ensemble mean
with the observation was improved to some extent. The NME and MAE of model ensemble mean was
closer to zero compared with the single model. Therefore, the multi-model ensemble mean was used
for further analysis.

**Table 3** Comparison between the simulated and observed monthly surface BC concentrations at Alert, Barrow, Tiksi,
and Zeppelin in 2010.

| Parameters | Sites | CAMchem | CHASER _re1 | CHASER _t106 | GEOS-Chem | GOCART –v5 | OsloCTM3 –v2 | Model ensemble mean |
|---|---|---|---|---|---|---|---|---|
| COR[a] | Alert | –0.24 | –0.22 | –0.15 | 0.35 | 0.20 | –0.24 | –0.10 |
| | Barrow | –0.28 | –0.08 | –0.06 | 0.06 | 0.00 | 0.01 | –0.06 |
| | Tiksi | –0.19 | 0.05 | –0.12 | 0.50 | 0.48 | 0.41 | 0.11 |
| | Zeppelin | 0.72 | 0.59 | 0.67 | 0.80 | 0.76 | 0.83 | 0.73 |
| NMB[b] | Alert | –86.75 | 115.06 | 100.97 | –57.81 | –34.31 | –92.38 | –9.21 |
| (%) | Barrow | –38.43 | 104.10 | 83.27 | –38.95 | –8.18 | –75.58 | 4.37 |
| | Tiksi | –82.03 | 10.31 | 12.90 | –69.76 | –67.34 | –84.82 | –46.79 |
| | Zeppelin | –79.93 | 142.64 | 120.44 | –45.57 | –9.81 | –75.98 | 8.63 |
| NME[c] | Alert | 86.75 | 151.30 | 147.94 | 66.37 | 70.77 | 92.38 | 74.69 |
| (%) | Barrow | 72.07 | 124.44 | 109.23 | 69.50 | 84.20 | 75.58 | 72.12 |
| | Tiksi | 82.03 | 64.55 | 72.45 | 70.16 | 68.82 | 84.82 | 60.81 |
| | Zeppelin | 79.93 | 142.64 | 120.44 | 45.57 | 38.59 | 75.98 | 42.06 |
| MB[d] | Alert | –29.03 | 11.08 | 8.74 | –21.41 | –16.07 | –30.16 | –12.81 |
| (ng m$^{-3}$) | Barrow | –22.13 | 15.35 | 10.77 | –19.10 | –11.12 | –30.40 | –9.44 |
| | Tiksi | –55.99 | –17.28 | –17.65 | –48.51 | –48.16 | –56.26 | –40.64 |
| | Zeppelin | –13.53 | 14.97 | 14.13 | –8.19 | –3.59 | –12.45 | –1.44 |
| MAE[e] | Alert | 29.03 | 31.05 | 31.55 | 22.85 | 22.23 | 30.16 | 23.56 |
| (ng m$^{-3}$) | Barrow | 29.01 | 28.91 | 27.82 | 26.71 | 30.30 | 30.40 | 25.22 |
| | Tiksi | 55.99 | 37.06 | 41.36 | 48.60 | 48.49 | 56.26 | 43.73 |
| | Zeppelin | 13.53 | 14.97 | 14.13 | 8.19 | 5.40 | 12.45 | 4.95 |

[a] Correlative coefficient. [b] Normalized mean bias. [c] Normalized mean error. [d] Mean bias. [e] Mean absolute error.





**3.2 Near-surface BC concentrations in the Arctic**
Before analyzing the responses of Arctic BC to emission reductions, it is necessary to understand the
spatial-temporal distribution of BC concentrations in the Arctic region. In this study, months from May
to October were defined as summer and November to April were defined as winter due to the special
geographical location of the Arctic (Aamaas et al., 2017).
The mean BC concentrations from the ensemble models near the Arctic surface (66–90°N) were
22.2 ng m$^{-3}$ during summer and 19.5 ng m$^{-3}$ during winter in 2010, respectively. Figure 2 shows that
the BC concentrations over the polar sea ice region in winter were much higher than that in summer.
The coverage of the polar dome expanded more southward in winter (Bozem et al., 2019; Law and
Stohl, 2007), allowing more BC from lower latitudinal regions to be transported into the Arctic.
Turbulent exchange and deposition were reduced during winter as the meteorological conditions in the
Arctic were stable and dry (Bradley et al., 1992; Bozem et al., 2019; Law and Stohl, 2007). In addition,
BC emissions in EAS, EUR, and RBU regions showed obvious monthly changes with higher emissions
from November to March as mentioned earlier (Section 2.1.1), leading to the relatively high BC
concentrations over the polar sea ice region in winter. Over the terrestrial areas within the Arctic Circle,
summer BC concentrations were higher than winter, especially in Siberia and Alaska, which were
attributed to intense BC emissions from biomass burning over these areas from Jun to Aug (Figure S3).

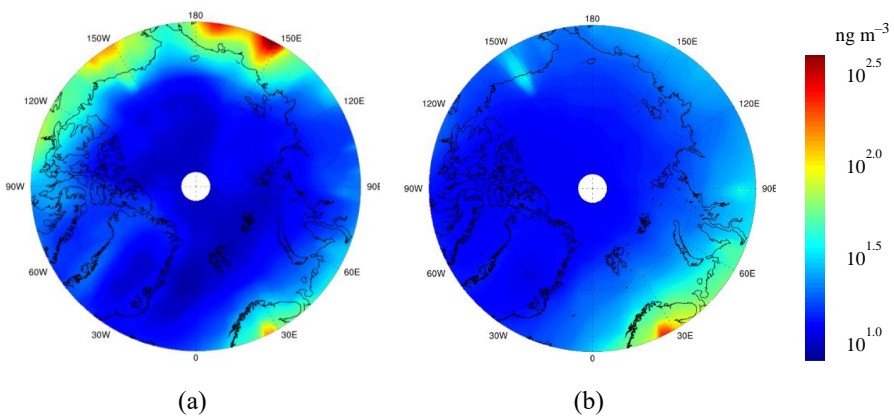

(a)                                (b)

**Figure 2.** Spatial distribution of near-surface BC concentrations in **(a)** summer and **(b)** winter in the Arctic in 2010.



### 3.3 Response of Arctic BC to 20% emission reductions

### 3.3.1 Contributions of regional emission reductions to the Arctic near-surface BC

The response of the Arctic near-surface BC to 20% emission reductions from different source regions was analyzed through emission perturbation simulations. Figure 3 shows the spatial distribution of the referred response above in summer and winter of 2010 based on multi-model ensemble mean results. The source region contributions to the surface BC concentrations exhibited significant seasonal variability with higher values in winter. The BC emission reductions in EAS almost affected the whole Arctic, especially in winter, indicating the significance of the intercontinental transport of BC. The spatial distribution of the Arctic near-surface BC response to SAS emission reductions was similar to that of EAS, but the extent was much weaker. The emission reductions from EUR, NAM, and RBU mainly affected the local and nearby areas, which was generally consistent with the spatial pattern of emissions (Figure 1). The contribution from MDE emission reductions was marginal.





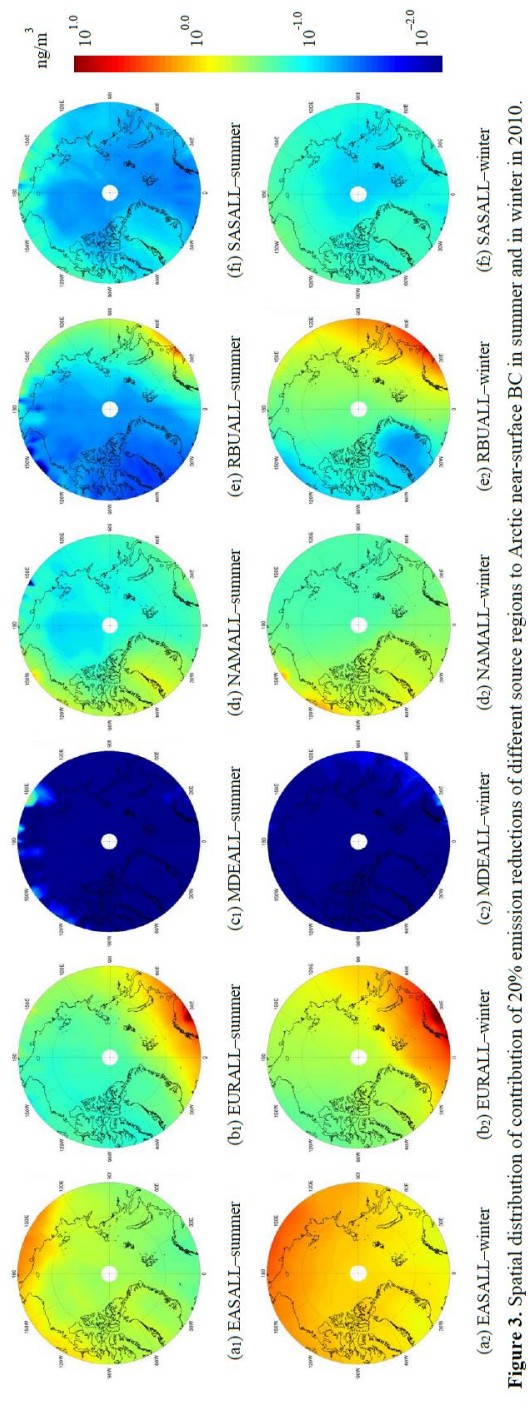

**Figure 3.** Spatial distribution of contribution of 20% emission reductions of different source regions to Arctic near-surface BC in summer and in winter in 2010.



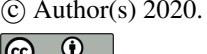



The monthly variations of the response of the Arctic near-surface BC concentrations to 20%
emission reductions from six source regions are presented in Figure 4. Results from the ensemble
simulations were averaged over the Arctic covering latitudinal areas from 66°N to the north pole. The
emission reductions from the total six source regions were 329.6 Gg in summer (May to October),
lower than that of 411.9 Gg in winter (November to April) (Table 1). Correspondingly, the contribution
from the six regions to the near-surface Arctic BC was 1.0–1.7 ng m$^{-3}$ in summer and 1.9–3.8 ng m$^{-3}$
in winter. Arctic sensitivities (Arctic concentration change per unit source region emission change) for
BC typically maximized from December to February for EUR and RBU and from March to May for
EAS and NAM (Shindell et al., 2008; AMAP, 2008). The enhanced sensitivity from December to May
resulted from faster transport and slower removal during winter as the meteorological conditions in
the Arctic were stable and dry (Law and Stohl, 2007). The results of deposition changes also proved
this result well (Figure S4). The wet deposition in summer was higher than that in winter, which was
3–10 times of dry depositions. Sharma et al. (2013) found that the Arctic region (north of 70°N) was
very dry during winter with an average daily precipitation rate between 0 and 1 mm day$^{-1}$. Precipitation
rates over some of the BC source regions such as Eurasia were at the same order of magnitude as the
Arctic. Less wet deposition and a shallow boundary layer resulted in higher BC concentrations near
the surface during winter. In the summertime, the Arctic region experienced 2 to 3 times higher
precipitation rates as well as wet depositions of BC relative to wintertime, thus resulting in lower
contributions to the near-surface BC concentrations.
The contribution of 20% BC emission reductions from EAS, EUR, MDE, NAM, RBU, and SAS to
the Arctic near-surface BC concentrations reached 0.88, 0.65, 0.01, 0.26, 0.29, and 0.11 ng m$^{-3}$,
respectively. Correspondingly, the reduced column BC loadings from the six regions above over the
Arctic was 8292.1, 2835.9, 28.8, 1774.6, 998.6, and 3381.1 ng m$^{-2}$, respectively.
The Arctic near-surface BC concentration response was found strongest to the 20% emission
reductions from EAS with the monthly contribution of 0.3–1.9 ng m$^{-3}$, accounting for 18.1%–51.4%
of the total reduced BC concentrations (Figure 4). On one hand, the BC emission reductions in EAS
were the largest among the six source regions (Table 1). On the other hand, BC emission reductions in
EAS can influence the Arctic lower troposphere via two pathways (Bozem et al., 2019; Stohl, 2006).
BC from northern regions of EAS can enter into the polar dome of the Arctic at low-level in winter, as



the air masses have cooled during transport. BC from eastern regions of EAS fast uplifted due to
convection and then followed by high altitude transport in northerly directions. Radiative cooling
eventually led to a slow descent into the polar dome area after air masses arrived in the high Arctic. It
occurred both in summer and winter. In addition to EAS, BC emission reduction from EUR also
showed significant impacts on the Arctic with the contribution of 0.3–1.2 ng m$^{-3}$, accounting for
20.1%–49.9% of the total reduced BC concentrations (Figure 4). Among the three regions in the Arctic
Circle (i.e. EUR, NAM, and RBU), EUR region had the largest BC emission reductions. Also, the
relatively short distance between EUR and the Arctic made EUR the second most important source
region to the Arctic. As for NAM and RBU, their 20% emission reductions induced moderate
reductions of the monthly Arctic near-surface BC concentrations by 0.1–0.4 and 0.1–0.7 ng m$^{-3}$,
respectively. The contribution of 20% emission reductions from SAS to the Arctic near-surface BC
concentrations was much lower of 0.1–0.2 ng m$^{-3}$ as a significant portion of BC originating from SAS
accumulated in the upper troposphere (Section 3.3.2). Compared to the five source regions discussed
above, the response of Arctic BC concentrations to emission reductions from MDE was negligible,
owing largely to the low emissions there and long distance from the Arctic.

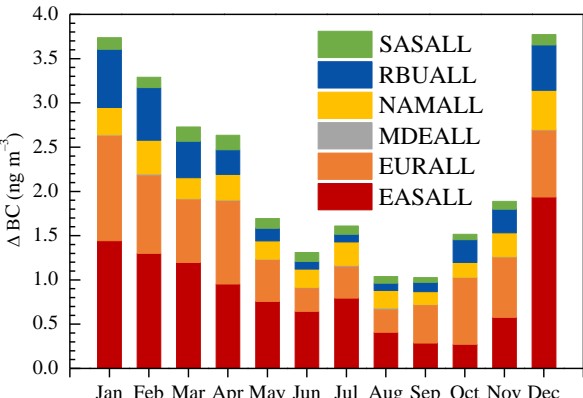

**Figure 4.** Monthly mean reduced concentrations of the near-surface Arctic BC due to 20% emission reductions of
six source regions in 2010.
**3.3.2 Contributions of regional emission reductions to the vertical BC profiles**
To assess the contributions from various source regions to the BC profiles based on the model
ensemble mean, the vertical stratification needed to be unified as most participating models had

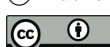



different vertical settings. Since CHASER had a relatively low coarse vertical resolution of 32 layers,
the other models were unified to the same vertical stratification, as detailed in Table S2.

As shown in Figure 5, the contributions of regional emission reductions to BC exhibited strong

vertical gradients over the Arctic. In general, the BC profiles displayed a bimodal pattern in summer,
showing peaks at around 1.0–1.6 km a.s.l. (4th and 5th layer) and 8.0–9.7 km a.s.l. (13th – 15th layer).
While in winter, the BC profiles showed a unimodal pattern with peaks around 0.6–1.0 km a.s.l. (3rd
and 4th layer). Long-range transport of air pollutions may occur near the planetary boundary layer
(Eckhardt et al., 2003; Stohl et al., 2002). High contributions in the low layers (e.g., 3rd – 5th layers)
were consistent with the height of the planetary boundary layer in the Arctic (Zhang, et al., 2018;
Cheng, 2011).

It has been summarized that there were several major transport pathways for BC into the Arctic

troposphere (Stohl, 2006). i) BC transported rapidly at low-level, followed by uplifting at the Arctic
front when it was located far north. Significant deposition of BC in the Arctic occurred mostly north
of 70°N via this transport route. This transport route derived often from the high BC emission areas in
northern EUR but seldom from the NAM and RBU. That was mainly due to that the BC emissions
existed at high enough latitudes in EUR, which can be north of the polar front. However, the BC
emissions in NAM and RBU were concentrated south of the polar front (Figure 1), thus BC emitted
from these two regions can't be easily transported into the Arctic through this pathway. ii) Cold air
masses into the polar dome transported at low-level. This pathway derived mainly from EUR and high-
latitude areas of EAS during winter. The contribution of 20% emission reductions from EUR to the
Arctic BC concentrations peaked at around 1.0 km a.s.l. with the multi-model ensemble mean value
of 0.5 ng m$^{-3}$ in summer, while peaked at lower altitude of around 0.6 km a.s.l. with the value of 0.9
ng m$^{-3}$ in winter. iii) BC could also ascend south of the Arctic followed either by high–altitude transport
or by several cycles of upward and downward transport, and finally slowly descended into the polar
dome due to radiative cooling. This was the frequent transport route from source regions such as NAM,
RBU, and east EAS. The contribution from NAM and RBU to the Arctic BC peaked at about 1.6 km
a.s.l. (0.2 ng m$^{-3}$) and 1.0 km a.s.l. (0.2 ng m$^{-3}$) in summer, and peaked at about 0.6 km a.s.l. (0.4 ng
m$^{-3}$) and 0.4 km a.s.l. (0.5 ng m$^{-3}$) in winter. The contribution from EAS including pathways ii and iii,
to the Arctic BC peaked at about 1.6 km a.s.l. (0.7 ng m$^{-3}$) in summer and peaked at about 1.6 km a.s.l.





(1.7 ng m$^{-3}$) in winter. The contribution from MDE was negligible.
As shown in Figure 5, BC can also be transported into the upper troposphere of the Arctic. Air
masses preferably kept their potential temperature almost constant during transport as the atmospheric
circulation can be well described by adiabatic motions in the absence of diabatic processes related to
clouds, radiation, and turbulence. The potential temperature was low within the polar dome area, and
thus only air masses experienced diabatic cooling were able to enter the polar dome (Stohl, 2006). That
is to say, the air masses from SAS and low-latitude regions of EAS were not easy to penetrate the polar
dome but can be lifted and transported to the Arctic in the middle and upper troposphere along the
isentropes (AMAP, 2011; Barrie, 1986; Law and Stohl, 2007; Stohl, 2006). This agreed well with the
previous study of Koch and Hansen (2005). The contribution from SAS to the Arctic BC concentrations
peaked at about 9.7 km a.s.l. (0.4 ng m$^{-3}$) in both summer and winter. This was also consistent with
the vertical profiles of BC shown in Stjern et al. (2016). The polar dome boundary was variable in time
and space and was not zonally symmetric. The range of polar dome expanded southward to about 40°N
over Eurasia in winter as the temperature difference of different latitudes became smaller (Bozem et
al., 2019; Law and Stohl, 2007), resulting in the contribution of EAS to the Arctic BC concentrations
in upper troposphere only peak in summer at 13$^{th}$ layer (8.0 km a.s.l.) with the value of 0.5 ng m$^{-3}$.

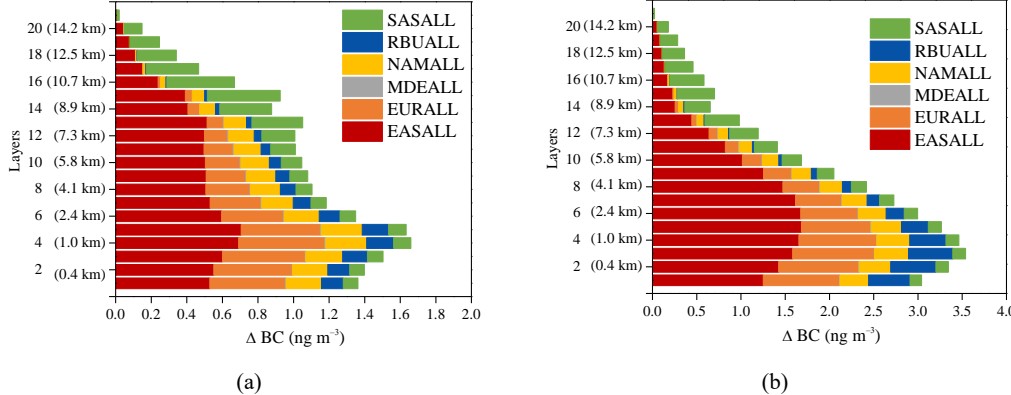

(a)              (b)

**Figure 5.** Contribution of 20% emission reductions of six source regions to BC concentrations in different vertical
layers **(a)** in summer and **(b)** in winter in the Arctic in 2010.
**3.3.3 Contributions of emission reductions to BC in different latitudinal bands**
To further analyze the response of the Arctic BC concentrations to emission reductions of six source



regions in HTAP2, the contribution of 20% emission reductions to BC concentrations at different
latitudes of the Arctic were calculated (Figures 6 and 7). In regard to the different horizontal resolution
of participating models, the Arctic region (66–90°N) was divided into eight latitudinal bands with a 3–
degree interval, which was based on the coarsest resolution of all models.
The response of the Arctic BC concentrations to emission reductions of six source regions became
weaker with the increase of the latitude due to the continuous loss of BC during transport (e.g., dry
and wet depositions) (Figure 6). The difference of contributions between two adjacent latitudinal bands
became smaller as closer to the north pole. The contributions of 20% emission reductions to the near-
surface Arctic BC concentrations were the highest between 66–69°N both in summer (2.3 ng m$^{-3}$) and
winter (4.6 ng m$^{-3}$), which were 1.3–2.6 times higher than the other latitudinal bands.
The contributions from EAS and EUR were higher than those from the other four regions in each
latitudinal band. In detail, the contributions from EUR (0.9 ng m$^{-3}$ in summer and 1.6 ng m$^{-3}$ in winter)
were higher than those from EAS (0.7 ng m$^{-3}$ in summer and 1.5 ng m$^{-3}$ in winter) in the latitudinal
band of 66–69°N as the near-surface BC concentrations there were more sensitive to the local emission
sources. In contrast, the contributions from EAS (0.4–0.6 ng m$^{-3}$ in summer and 1.1–1.3 ng m$^{-3}$ in
winter) were higher than those from EUR (0.2–0.6 ng m$^{-3}$ in summer and 0.5–1.0 ng m$^{-3}$ in winter) in
the other high latitudinal bands where long-range transport played the dominant role.

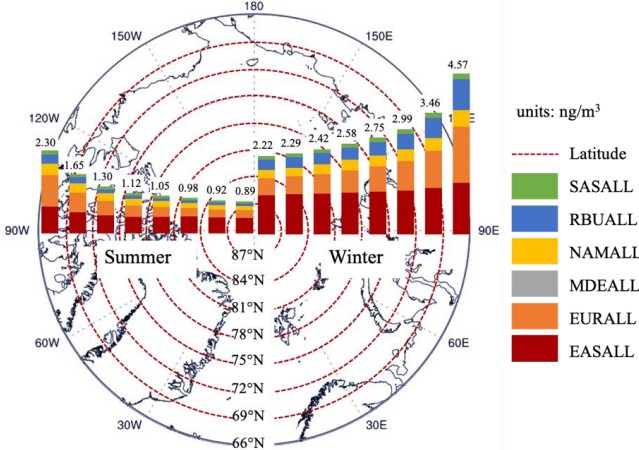


**Figure 6.** Contributions of 20% emission reductions of different regions to near-surface BC concentrations in each
latitudinal band of the Arctic. The results of summer and winter correspond to the left and right panel in the figure.





Figure 7 further depicts the response of the vertical Arctic BC concentrations to 20% emission
reductions. The contributions of eight latitudinal bands showed a typical bimodal pattern in summer
with peaks at 0.6–1.6 km a.s.l. (3rd – 5th layer) and 8.0–8.9 km a.s.l. (13th and 14th layer), while the
contribution displayed a single peak at the 0.4–1.0 km a.s.l. (2nd – 4th layer) in winter. Similar to section
3.3.2, the peak value of the contribution at the low layer was due to the transport of EAS, EUR, NAM,
and RBU emission reductions to the Arctic through different pathways both in summer and winter. The
peak value in the high layer in summer was due to the transport of EAS and SAS. However, a high
contribution of 20% emission reductions to BC concentrations in SAS was found in the high layer,
while the contribution was low in other regions, leading to a single peak in winter. The statistical results
of SAS indicated that the contribution in vertical appeared one peak at the 15th layer (9.7 km a.s.l.)
with a value of 0.40–0.44 ng m$^{-3}$ both in summer and winter (Figure S5).

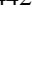

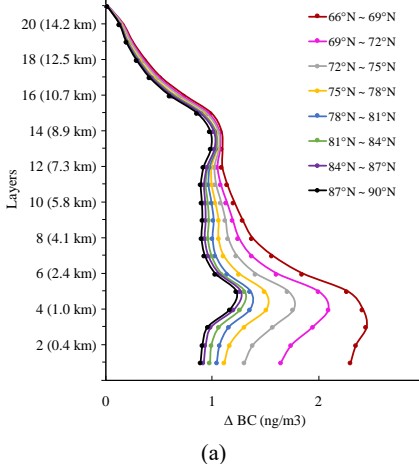

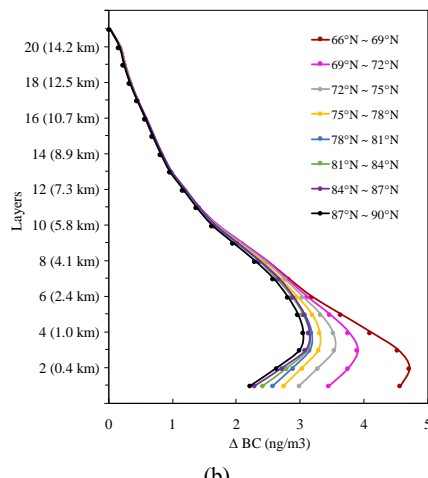

(a)                                              (b)

**Figure 7.** Contributions of 20% emission reductions of all six source regions to the vertical BC concentrations of the
Arctic in different latitude bands varies with vertical layers in **(a)** summer and **(b)** winter in 2010.

As same as the whole Arctic region (section 3.3), the contributions of 20% emission reductions to
BC concentrations in eight latitude bands were higher in winter than in summer, whether near-surface
or in vertical. The total contribution of six source regions to BC concentrations in eight latitude bands
of the near-surface Arctic was 0.9–2.3 ng m$^{-3}$ in summer and 2.2–4.6 ng m$^{-3}$ in winter, respectively
(Figure 6). The peak of total contribution in vertical at the lower layer was 1.2–2.5 ng m$^{-3}$ in summer,
and 2.9–4.7 ng m$^{-3}$ in winter, respectively (Figure 7).





**3.4 Benefit of BC emission reductions on the decrease of Arctic temperature**
The impact of BC emission reductions on decreasing the Arctic (60–90°N) surface temperature was
assessed by using ARTP (See methods in Section 2.2). Aerosol effects, BC deposition on snow, and
BC semi-direct were considered in the calculation of ARTP (Aamaas et al., 2017). As shown in Figure
8, the response of Arctic temperature to emission reductions was the most significant at the time scale
of 10 years and then gradually decreased with the passage of time. For each source region, the Arctic
temperature response was significantly higher in winter than in summer as ARTP was seasonal
dependent with higher values in the warm seasons. Obviously, the Arctic benefited the most from
emission reduction from EAS with more than 300 and 660 μK decreases in summer and winter after
10 years, respectively. The influences of SAS and EUR emission reductions on the temperature
decrease in winter were similar, reaching about 10–320 μK and 14–370 μK after 10-100 years,
respectively. However, in summer, the influence of SAS on temperature decrease (8–230 μK) was
more than twice that of EUR (3–90 μK). Although the ARTP of EUR was higher than that of SAS, the
difference between emission reductions in the two regions was more obvious in summer (Table 1). The
temperature response to NAM in summer was similar to that from EUR, while this effect was weaker
than that from EUR in winter. This was mainly due to the smaller ARTP of NAM than EUR although
larger emission reductions were found from NAM. The temperature response to RBU was small due
to the low BC emission reductions even a high ARTP was associated with RBU. The minimum
temperature response was found from MDE due to the least emission reductions and small ARTP. In
spite of the higher Arctic temperature response to EAS than SAS in the target year of this study, a
number of studies have shown that BC emissions in South Asia were increasing in recent years (Sahu
et al., 2008; Paliwal et al., 2016; Sharma et al., 2019) while the emissions of East Asia were exhibiting
a downward trend especially from China (Chen et. al., 2016), thus it should be given more attention to
the impact assessment of South Asia on the Arctic in the future.





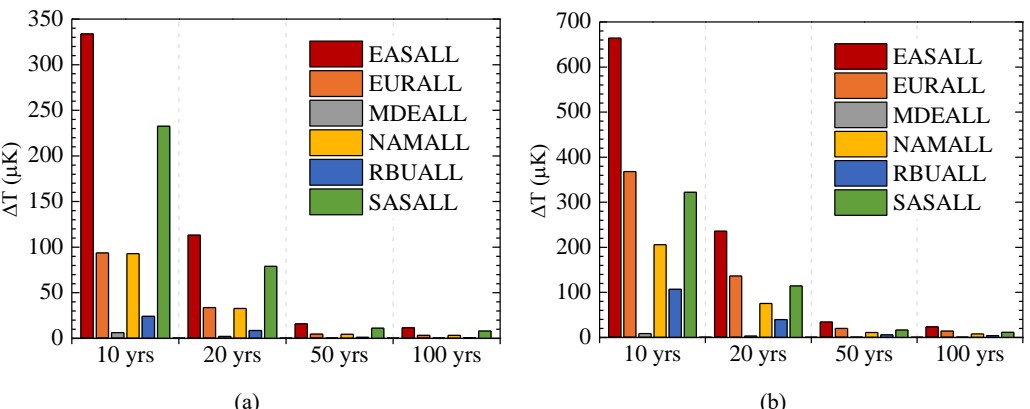

(a)          (b)

**Figure 8.** Arctic temperature response to 20% regional BC emission reductions in **(a)** summer and **(b)** winter after 10, 20, 50, and 100 years.

In addition, the impacts of six source regions on the Arctic and global temperature were compared in this study. As shown in Figure 9, the Arctic and global temperature response to BC emission reductions from the six source regions ranged from about 27–780 μK and 10–290 μK in summer after 10, 20, 50, and 100 years, respectively, and they were about 61–1675 μK and 16–470 μK, respectively in winter. It can be seen that the difference of the temperature response between the Arctic and the globe was more obvious in winter. Overall, the Arctic temperature response was more sensitive to the whole globe in regard to the same emissions perturbation.

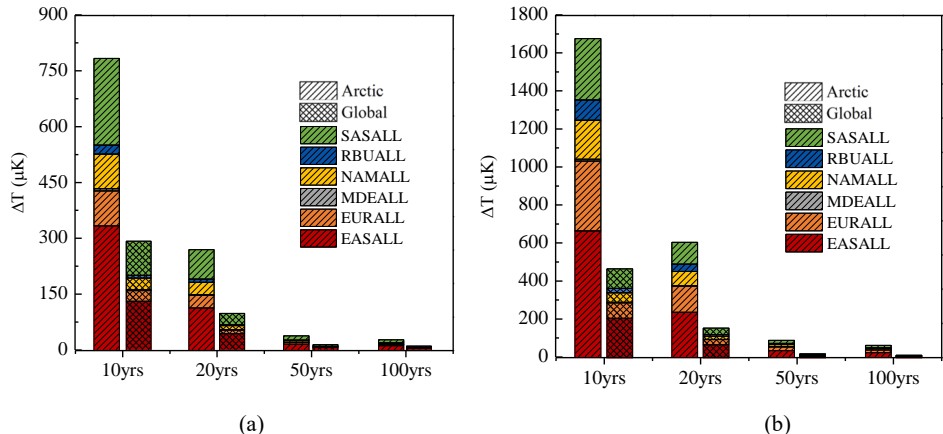

(a)          (b)

**Figure 9.** Global and Arctic temperature responses to 20% regional BC emission reductions in **(a)** summer and **(b)** winter after 10, 20, 50, and 100 years.

It should be noted the estimation of temperature response was subject to large uncertainties for the



following reasons. On the one hand, even though the HTAP2 emissions database were all constructed
by bottom-up methods, the different inventories and spatiotemporal distributions were constructed
with sub-regional (country, state, county or province level) activity data and emission factors, which
lead to inconsistencies at the borders between two adjacent inventories. The version 5 of Evaluating
the Climate and Air Quality Impacts of Short-Lived Pollutants (ECLIPSEv5, http://eclipse.nilu.no)
estimated a 2010 emission inventory, that serves also as a reference point for all projections (Janssens-
Maenhout et al., 2015). At the global level, a relatively good agreement was found with small relative
emission differences compared with the ECLIPSEv5 emission inventory for the aggregated sectors in
2010. However, larger differences of 29% between HTAP2 and ECLIPSEv5 emissions was present for
BC since ECLIPSEv5 relied on provincial statistics for China which resulted from higher coal
consumption than reported national statistics.

On the other hand, the time evolution of $R_T$, a parameter in the calculation of ARTP was also one

factor causing the uncertainty of temperature response calculation. This impulse response function was
only based on one coupled atmosphere-ocean climate model GISS-ER in this study, while Olivié and
Peters (2013) have found a spread in the GTP (20) value of BC of about −60 to +80% due to variability
of $R_T$ among various models. However, the uncertainty in $R_T$ was less relevant for the regional patterns.
Forcing-response coefficients didn't exist on a seasonal basis since emissions occurring during
Northern Hemisphere summer and winter season were differentiated (Aamaas et al, 2017). Hence, the
seasonal differences presented here in the ARTP values were not due to potential differences in the
response sensitivities, but due to differences in the RF. The temperature response will vary by species
and location, such as between land surface and ocean surface. These differences are not accounted for
in this study, but the increased efficacy in the RCS matrix towards the NH can be partly attributed to a
larger land area fraction in the NH (Shindell et al., 2015). Besides, recent studies have found that the
positive radiation budget of BC has been largely compensated by rapid atmospheric adjustment, this
means that the responses of surface temperatures to BC could be weaker than expected (Stjern et al.,
2017; Takemura and Suzuki, 2019).

Although the HTAP2 emissions database contain uncertainties and ARTP calculations are

simplifications, these emission metrics are useful, simple, and quick approximations for calculating
the temperature response in the different latitude bands for emissions of BC.



## 4. Conclusions


The CAMchem, CHASER_re1, CHASER_t106, GEOS-Chem, GOCART, and Oslo CTM3 in
HTAP2 experiment were used in this study to estimate the responses of Arctic BC to multi-region
emission reductions in 2010. Six regions (e.g., EAS, EUR, MDE, RBU, NAM, and SAS) were selected
as the source regions and the Arctic was the receptor region. HTAP2 set up the base scenario with all
BC emissions, and also simulated BC concentrations with 20% reduction of anthropogenic emissions.
The AGPT was further used to calculate the benefit of BC emission reductions on the decrease of
Arctic temperature.
The statistical results of 20% BC emission reductions showed that emission reductions in EAS were
the largest with the values of 355.6 Gg yr$^{-1}$, followed by SAS (232.5 Gg yr$^{-1}$), EUR (65.3 Gg yr$^{-1}$),
NAM (62.2 Gg yr$^{-1}$), RBU (18.6 Gg yr$^{-1}$), and MDE (5.3 Gg yr$^{-1}$). The BC emission reductions in the
EAS, EUR, and RBU were higher from November to March.
The temporal variations of simulations from different models were relatively consistent as the
correlations of the simulated BC concentrations among different models ranged from 0.33 to 0.98.
However, the simulated BC concentrations didn't agree so well with observations at monitoring sites
except Zeppelin. The consistency of the model ensemble mean value with the observation was
significantly improved and the results were acceptable to use for further analysis.
The contribution of 20% BC emission reductions from EAS, EUR, MDE, NAM, RBU, and SAS to
the Arctic near-surface BC concentrations reached 0.88, 0.65, 0.01, 0.26, 0.29, and 0.11 ng m$^{-3}$,
respectively. Correspondingly, the reduced column BC loadings from the six regions above over the
Arctic was 8292.1, 2835.9, 28.8, 1774.6, 998.6, and 3381.1 ng m$^{-2}$, respectively.
BC emission reduction from EAS and EUR showed significant impacts on the near-surface Arctic
with the contribution of 0.3–1.9 ng m$^{-3}$ and 0.3–1.2 ng m$^{-3}$, accounting for 18.1%–51.4% and 20.1%–
49.9% of the total reduced BC concentrations, respectively. The BC profiles displayed a bimodal
pattern in summer with peaks at around 1.0–1.6 km a.s.l. (4$^{th}$ and 5$^{th}$ layer) and 8.0–9.7 km a.s.l. (13$^{th}$
– 15$^{th}$ layer). While the BC profiles showed a unimodal pattern with peaks around 0.6–1.0 km a.s.l.
(3$^{rd}$ and 4$^{th}$ layer) in winter.
The response of Arctic BC to emission reductions from source regions in winter was higher than





that in summer. The contributions of 20% emission reductions to the near-surface Arctic BC
concentrations were the highest between 66–69°N both in summer (2.3 ng m$^{-3}$) and winter (4.6 ng m$^{-}$
$^{3}$), and became weaker with the increase of the latitude.
The response of Arctic temperature to BC emission reductions was the most significant at the time
scale of 10 years and then gradually decreased with the passage of time. The Arctic had benefited the
most from emission reduction in EAS with more than 300 and 660 μK decreases in summer and winter
after 10 years, respectively. The Arctic temperature response was more sensitive to the whole globe in
regard of the same emissions perturbation. The estimation of temperature response was subject to large
uncertainties due to the uncertainties in the calculation of ARTP and emissions of BC in source regions.
Overall, this study provided insights on the source regions and seasonal contributions of Arctic BC
from the most recent international ensemble modeling efforts. The discrepancy between model results
and observations and the spread among different HTAP models may be attributed to various factors
such as emissions in the remote Arctic, physical parameterizations, and convection and deposition
processes. This would subsequently result in large uncertainties of the climatic effects of air pollutants.
More observation sites on the typical transport pathways from sources regions to the Arctic should be
planned to improve the model capability of simulating the transport behavior of black carbon.
**Data availability**
All data used in this study can be obtained through the AeroCom servers and web interfaces,
accessible at http://aerocom.met.no.

**Author contributions**
KH and JSF designed this study. ML, KS, DH, TK, MC, and ST performed modeling. NZ analyzed
data and wrote the paper. All have commented on and reviewed the paper.



**Competing interests.**
The authors declare that they have no conflict of interest.

**Acknowledgements**
We sincerely thank for the HTAPv2 international initiative. This work was partially supported by
the National Natural Science Foundation of Shanghai (18230722600), the National Key R&D Program
of China (2018YFC0213105), and the National Natural Science Foundation of China (91644105).



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
