# Peer review of "Responses of Arctic Black Carbon and Surface Temperature to Multi-Region Emission Reductions: an HTAP2 Ensemble Modeling Study"

_Atmospheric Chemistry and Physics, 2020_

## Referee Comment (RC1) · Anonymous Referee #1 · 14 Feb 2021

The manuscript by Zhao et al. analyzed model simulation results from the 2nd phase of The Task Force Hemispheric Transport of Air Pollution (HTAP2). Six global models were chosen and model ensemble results are used in the analysis. The contributions of 20% emission reductions from six source regions to the Arctic region are focused, including surface concentrations, vertical profiles, and temperature response. The focus on Arctic based on the HTAP international initiative is a useful addition to AMAP. Overall, this paper is clearly structured with methodology sound and generally well written. However, the following comments should be addressed before the acceptance of the manuscript.

[Figure]

Major comments: 1. Abstract: The abstract is not very clearly written. Line 35: The abbreviation "EAS" should be deleted. Are the percentages monthly contributions? Line 36: "Russia Belarus Ukraine" is better changed to Russia/Belarus/Ukraine. Line 38-39: It is not clear how different transport pathways affect the vertical profiles. Overall, the abstract is suggested to be more informative.

2. Line 281 – 282 shows higher BC concentrations in summer than in winter. While in Section 3.3.1 (Line 300 - 301), source region contributions to BC showed higher concentrations in winter. This seems contradictory to me and please explain in details.

3. Line 419 – 425: In Figure 6, it seems that the contribution from East Asia to BC at different latitudes remained almost constant while that from Europe decreased obviously from lower latitudes to the Arctic pole. Please double check and make explanations when applicable.

Other comments: 1. Section 2.1: What's the data source of European emissions?

2. Figure 1: the unit of Figure 1 doesn't show correctly. The labels of (a) – (g) are missing in the figure.

3. Table 1: The 20% emission reductions in January are missing in the table.

4. Line 142: change "indicated" to "showed"

5. Line 158: add "spatial" in front of "distribution"

6. Line 161- 162: Please state why the six models are chosen since more models have been used in HTAP

7. Line 163: change "of the" to "from"

8. Line 249: change "temporal" to "monthly"

9. Line 254: Stohl et al., 2013 also showed that appropriate temporal allocation of the residential emissions can improve the model simulation results.

10. Line 269 – 270: More discussions on the performance of model ensemble results should be added.

11. Line 282-283: It is hard to see "BC concentrations over the polar sea ice region in winter were much higher than that in summer." from the contours in Figure 2.

12. Line 328-331: Are these values annual average or seasonal average?

13. Line 332: The sentence is suggested to rephrase as "The response of Arctic near-surface BC concentration……"

14. Line 337: Do you mean "at low altitudes"? And what altitudes?

15. Line 359: delete "low"

16. Line 431: change "concentrations" to "profiles"

17. Line 446: "section 3.3" may be wrong

18. Line 450: It is not clearly what does lower layer mean here

19. Line 463-465: It is not clearly stated about the difference of temperature response between EUR and SAS.

20. Line 485-486: This sentence is misleading. Do you mean the Arctic temperature response was more sensitive than the global temperature response?

---

## Referee Comment (RC2) · Anonymous Referee #2 · 22 Feb 2021

This study investigated the responses of Arctic BC concentrations and surface temperature to 20% anthropogenic emission reductions from six source regions (East Asia, Europe, Middle East, North America, Russia-Belarus-Ukraine, and South Asia) by using several global aerosol models. The contribution from East Asia was estimated to be the largest (18.1-54.1%) among the six source regions. The authors also showed that the source contributions of the Arctic BC vary with altitude and latitude and the responses of the Arctic BC to emission reductions from the six source regions decrease with latitude. Finally, this study indicated that the reductions of global BC emissions are important for the Arctic climate by showing that the response of the Arctic air temperature to BC reduction is larger than that of the global air temperature.

[Figure]

The results of this study are useful for understanding the current status of the estimation of the source contributions of Arctic BC by global aerosol models. However, as shown below, there are some important problems that need to be clearly described.

Major comments:

1)

This study uses six global models. Two models (CHASER) overestimate the BC concentrations in the Arctic by a factor of two, and four models (CAMchem, GEOS-chem, GOCART, and OsloCTM3) underestimate BC by more than 50%. The authors use the averages of these six model results and describe that the averages reproduce the observed BC concentrations in the Arctic well (lines 268-271). However, none of these models can simulate Arctic BC well. It is not meaningful to show as if the agreement with observations was improved by averaging the multiple model results that cannot reproduce the Arctic BC. The authors need to describe more clearly why it makes sense to use the averages of the model results that cannot reproduce the observations well.

2)

As described in the comment (1), two models (CHASER) overestimate Arctic BC by a factor of 2 and four models underestimate it by more than 50%. Therefore, the two CHASERs have about 4 times higher concentrations than the other four models. Then, the simple averages of these models strongly depend on the results of the two CHASERs with their contributions of about 2/3 to the total (CAMchem: CHASER_re1: CHASER_t106: CEOS-chem: GOCART: OsloCTM3 = 1: 4: 4: 1: 1: 1). If this is correct, the multi-model averages shown in this study will not be very different from the results of CHASER (the contributions of the other models will be too small).

Considering these points, I suggest the authors to revise the manuscript for the three points below. First, please discuss the spatial distributions and source contributions of BC in the Arctic for each model. Second, please use one CHASER, not two. Third,

please consider using medians instead of averages, as has been done in other multi-model studies.

3)

Section 3.3.2 discusses the vertical profiles of Arctic BC. However, they are not validated by observations. Because several aircraft measurements such as HIPPO, ATom, and ARCTAS are available, the authors should use such data to validate the simulated vertical profiles of Arctic BC.

4)

Several observational and trajectory-based studies have shown that the contribution from Asia (especially South Asia) to the near-surface BC concentration in the Arctic is limited (e.g., Stohl, 2006; Matsui et al., 2011). Please clarify whether the results of this study are consistent with the findings of these previous studies.

Stohl (2006): doi:10.1029/2005JD006888, Matsui (2011): doi:10.1029/2010JD015067

Specific comments:

5) Lines 35 and 37

18.1 – 54.1%: Please clarify the meaning of the range.

6) Lines 118 – 119

Please add the amount of global BC emission flux from anthropogenic sources and describe how different it is from the CMIP6 emissions.

7) Lines 141 – 144, Table 1

Please describe how the authors determined the monthly variations of BC emissions. Please add the values of global emissions.

8) Line 209

What is "m" in equation (1).

9) Line 314

1.0 – 1.7 ng m-3, 1.9 – 3.8 ng m-3: Please clarify the meaning of the ranges.

10) Line 329

The sum of the six sources (from 20% emission reductions) is about 2 ng m-3. I suspect that the contributions from sources other than the six regions (e.g., anthropogenic sources in the Arctic, biomass burning sources) are large.

11) Line 333

0.3 – 1.9 ng m-3, 18.1 – 51.4%: Please clarify the meaning of the ranges. Please show the contributions of anthropogenic and biomass burning sources other than the six sources. Are the contributions of the six sources to total BC in the Arctic dominant?

12) Lines 342 – 343

Please clarify the meaning of the ranges.

13) Line 356

Section 3.3.2: the vertical profiles of BC should be validated by the available aircraft measurements.

14) Line 433

How important are these two peaks? Is this consistent with observations?

15) Lines 449 – 451

Please clarify how important these concentrations are.

Please show the contributions of these sources to total BC.

16) Line 457

10 years: What does this mean? The time scale of BC impacts on Arctic temperatures is probably much shorter than 10 years.

17) Lines 460 – 461

What do these temperatures mean? It's unclear how they correspond to actual temperature changes.

18) Line 475

The authors conclude that South Asia will be important. Is this consistent with previous studies (e.g., Stohl, 2006)?

19) Line 522

Are the two CHASERs different enough to be considered as different models?

20) Line 534

Is the correlation coefficient of 0.98 due to the two CHASERs? I think the value is too high as a correlation between independent models.

21) Lines 536 – 537

I don't think it makes sense to show as if the agreement with observations was improved by averaging the multiple models that overestimate and underestimate observations. None of the models in this study can simulate Arctic BC reasonably well.

22) Line 543

Please clarify the meaning of the range.

23) Lines 545 – 547

Is this consistent with observations?

---

## Author Comment (AC1) · 7 Apr 2021

**Response to Reviewer #1's Comments**

Anonymous Referee #1:

The manuscript by Zhao et al. analyzed model simulation results from the 2nd phase of The Task Force Hemispheric Transport of Air Pollution (HTAP2). Six global models were chosen and model ensemble results are used in the analysis. The contributions of 20% emission reductions from six source regions to the Arctic region are focused, including surface concentrations, vertical profiles, and temperature response. The focus on Arctic based on the HTAP international initiative is a useful addition to AMAP. Over- all, this paper is clearly structured with methodology sound and generally well written. However, the following comments should be addressed before the acceptance of the manuscript.

We sincerely thank for the reviewer's in-depth comments and helpful suggestions on this manuscript. Based on the specific comments, we have responded to all the comments point-by-point and made corresponding changes in the manuscript as highlighted in red color. The reviewer has raised a number of issues and we quite agree. We feel the substantial revisions based on the reviewer's comments have greatly improved the quality of this manuscript. Please check the detailed responses to all the comments as below.

Major comments:

1. Abstract: The abstract is not very clearly written. Line 35: The abbreviation "EAS" should be deleted. Are the percentages monthly contributions? Line 36: "Russia Belarus Ukraine" is better changed to Russia/Belarus/Ukraine. Line 38-39: It is not clear how different transport pathways affect the vertical profiles. Overall, the abstract is suggested to be more informative.

Response: Thanks for the suggestion. In the revised abstract, the term "East Asia" appeared several times, thus we still keep this abbreviation.

The percentages (18.1%–51.4%) mean the ratios between the response of Arctic near-surface BC concentrations to emission reductions from EAS versus that from all six regions. To avoid confusion, we changed percentages to mass concentrations.

As for the term "Russia Belarus Ukraine", it has been changed to "Russia-Belarus-Ukraine" according to the protocol of HTAP.

The emission reductions from NAM, RBU, EUR, and EAS mainly influenced the BC concentrations in lower troposphere of the Arctic, while most of the BC in upper troposphere of the Arctic was transported from SAS. We have added this information in the revised section "ABSTRACT" (lines 34 – 42).

2. Line 281 – 282 shows higher BC concentrations in summer than in winter. While in Section 3.3.1 (Line 300 - 301), source region contributions to BC showed higher concentrations in winter. This seems contradictory to me and please explain in details.

Response: Thanks for the comment. We have checked the data and confirmed the accuracy of the results. Spatial distribution of emission inventory showed the emissions from Russia and North America exhibited high BC intensities in June, July, and August (Figure S3). These hotspots were dominated by biomass burning near the Arctic circle, explaining the higher BC concentrations in summer than in winter. Matsui (2011) compared the response of BC over the North American Arctic to biomass burning (BB) in Russia (Russian BB) and anthropogenic emissions (AN) in East Asia (Asian AN) in 2008. They demonstrated that Russian BB emissions in summer (June – July) were more efficiently transported to the Arctic than Asian AN. In the calculation of source region contributions to BC in the Arctic in this study, HTAP2 focused on the anthropogenic emissions. The BC emission reductions from EAS, EUR, and RBU showed significant monthly variations with higher values from November to March. In addition, BC from source regions were easier to be transported to the Arctic in winter due to the seasonal variation of polar dome. The seasonal variation of emissions and transport mechanism led to the greater contribution from anthropogenic emissions in winter. Hence, there is no contradiction between Line 281 – 282 and Line 300 – 301.

In the revision, we have deleted the simulation results of CHASER_t106 and rearranged the results. The mean BC concentrations from the ensemble models near the surface Arctic (66–90°N) were 18.6 ng m$^{-3}$ during summer and 16.6 ng m$^{-3}$ during winter in 2010, respectively. Please see lines 306 – 309 for the changes.

[Figure]

**Figure S3**. Spatial distribution of monthly BC emissions in 2010.

3. Line 419 – 425: In Figure 6, it seems that the contribution from East Asia to BC at different latitudes remained almost constant while that from Europe decreased obviously from lower latitudes to the Arctic pole. Please double check and make explanations when applicable.

Response: Figure 6 showed the response of the Arctic BC concentrations to emission reductions from six source regions became weaker with the increase of the latitude. The downward trends of the response of BC in the Arctic near surface to emission reductions from EUR and RBU were more obvious than that of other regions.

We further extracted the simulation results of BC dry and wet depositions (Figure S7). The results showed that the maximum values of BC dry and wet depositions caused by the emission reductions from EAS, EUR, NAM, RBU, and SAS occurred at 30°N–40°N, 50°N, 40°N, 50°N–60°N, and 20°N–30°N, respectively, which was consistent with the latitude of each region. BC dry and wet deposition decreased with the increase of transport distance, and the decreasing rates became slow. The changes of dry and wet depositions caused by emission reductions from EUR and RBU were still obvious in the Arctic region (66°N–90°N), while depositions caused by emission reductions from the other regions tended to be gentle (Figure S7). This explains why the contribution from EAS to BC at different latitudes remained almost constant while that from EUR decreased obviously from lower latitudes to the Arctic pole.

We have added more discussions lines 465 – 472 in the revision.

[Figure]

(a) dry deposition – summer

(b) dry deposition – winter

(c) wet deposition – summer

(d) wet deposition – winter

**Figure S7**. Reduced dry and wet depositions of the near surface Northern Hemisphere BC due to 20% emission reductions from six source regions during summer and winter in 2010.

Other comments:

1. Section 2.1: What's the data source of European emissions?

Response: European emissions were obtained from European Monitoring and Evaluation Programme (EMEP) and Netherlands Organisation for Applied Scientific Research (TNO). We have added this information. It is added in lines 117 – 122 in the revision.

2. Figure 1: the unit of Figure 1 doesn't show correctly. The labels of (a) – (g) are missing in the figure.

Response: The unit of Figure 1 has been changed to "$kg\ m^{-2}\ s^{-1}$" and the labels of (a) – (g) are added. Please check the corrected figure.

3. Table 1: The 20% emission reductions in January are missing in the table.

Response: The 20% emission reductions in January have been added in Table 1. Please check the updated Table 1.

4. Line 142: change "indicated" to "showed"

Response: We have made the change.

5. Line 158: add "spatial" in front of "distribution"

Response: We have made the change.

6. Line 161- 162: Please state why the six models are chosen since more models have been used in HTAP

Response: Among the participating models in HTAP2, C-IFS, C-IFS_v2, CAMchem, CHASER_re1, CHASER_t106, EMEP_rv48, GEOS5, GEOSCHEMADJOINT, GOCART-v5, OsloCTM3-v2, HCMAQ, and SPRINTARS simulated the mixing ratio of BC in 2010. However, C-IFS, C-IFS_v2, and CHASER_t106 didn't conduct experiments of emission reductions from MDE and RBU, while GEOS5 didn't conduct experiments of emission reductions from MDE and NAM. HCMAQ only simulated BC in Apr., Jan., Jul., and Oct. of 2010. The dry and wet deposition data simulated by EMEP_rv48 were not found in the HTAP2 database. As for SPRINTARS, we thought there were some mistakes in the emission inventory of scenario simulations. We have stated this clearly in lines 166 – 169.

7. Line 163: change "of the" to "from"

Response: We have made the change.

8. Line 249: change "temporal" to "monthly"

Response: We have made the change.

9. Line 254: Stohl et al., 2013 also showed that appropriate temporal allocation of the residential emissions can improve the model simulation results.

Response: We have added this reference in lines 260 – 262.

10. Line 269 – 270: More discussions on the performance of model ensemble results should be added.

Response: The vertical profiles of simulated BC concentrations of the BASE simulation were also compared with aircraft measurements from HIAPER Pole-to-Pole Observations (HIPPO) during 24 March–16 April 2010 (Figure S2b). Different from comparison between observed and simulated BC concentrations near the surface, the vertical profiles of BC concentrations were overestimated by most models. As the aircraft ascended and descended along each flight track, BC concentrations from HIPPO varied with time, latitude, longitude, and altitude. However, most of the simulation results of HTAP2 were provided in the temporal resolution of monthly, simulation and observation results cannot be exactly matched. This partly explained the difference between the simulations and observations. In general, the model ensemble mean could capture the vertical pattern of observed BC profiles.

[Figure]

**Figure S2 (b)** Comparison between the vertical profiles of simulated (BASE simulation) and observed BC concentrations (HIPPO) during 24 March–16 April 2010.

No single model could reproduce the BC concentrations in the Arctic well, and models performed differently at different monitoring sites. Compared with the observations, the best correlation (0.83) was found at Zeppelin from Oslo CTM3, while the smallest NMB (38.59%) and MAE (5.40 ng m$^{-3}$) were found at Zeppelin from GOCART. Moreover, limited by the monitoring data, this paper only compared the observations and simulations at four sites. If data of more monitoring sites can be obtained, the difference of performance of these models at different monitoring sites would be more obvious. Therefore, the multi-model ensemble mean was used for further analysis.

We have added more discussions on the performance of model ensemble results. Please see lines 278 – 287 in the revision.

11. Line 282-283: It is hard to see "BC concentrations over the polar sea ice region in winter were much higher than that in summer." from the contours in Figure 2.

Response: We have changed "much higher" to "higher". Figure 2 shows that the filling color of the ocean is greener in summer and yellower in winter. Please see the newly plotted Figure 2.

12. Line 328-331: Are these values annual average or seasonal average?

Response: The values were annual contribution. We have made the change. Please see the change in lines 357 – 360.

13. Line 332: The sentence is suggested to rephrase as "The response of Arctic near-surface BC concentration. . .. . ."

Response: We have made the change.

14. Line 337: Do you mean "at low altitudes"? And what altitudes?

Response: The high Arctic lower troposphere in general is quite well isolated from the rest of the Arctic by a transport barrier referred to as the polar dome. The polar dome is formed by sloping isentropes and varies with space and time. The placement of the polar dome is more typical of the winter/spring situation, whereas in summer the dome is much smaller. Also note that the dome is not homogeneous but is itself highly stratified with strong vertical gradients. We have deleted the "at low-level" in the revision.

15. Line 359: delete "low"

Response: We have made the change.

16. Line 431: change "concentrations" to "profiles"

Response: We have made the change.

17. Line 446: "section 3.3" may be wrong

Response: We have changed "section 3.3" to "Section 3.3.1 & 3.3.2".

18. Line 450: It is not clearly what does lower layer mean here

Response: We have changed "lower layer" to "around 0.6–1.6 km a.s.l. ($3^{rd}$ – $5^{th}$ layers)" in lines 496 – 497.

19. Line 463-465: It is not clearly stated about the difference of temperature response between EUR and SAS.

Response: In summer, the response of the temperature decrease to emission reductions from EUR was 3–90 µK after 10-100 years, while the response of the temperature decrease to emission reductions from SAS was 8–230 µK after 10-100 years. 8–230 µK were more than twice as much as 3–90 µK. The temperature response was proportional to both ARTP and BC emission reductions. The ARTP of SAS was lower than that of EUR. However, the BC emission reductions from SAS were much higher than that from EUR and the difference between emission reductions from the two regions was more obvious in summer. BC emission reductions from SAS in summer was about five times that of BC emission reductions from EUR. This led to a higher temperature response to emission

reductions from SAS than emission reductions from EUR. We have stated more clearly about this in lines 515 – 520.

20. Line 485-486: This sentence is misleading. Do you mean the Arctic temperature response was more sensitive than the global temperature response?

Response: As you said, our original intention was to express that the Arctic temperature response was more sensitive than the global temperature response. We have corrected the grammatical errors in lines 536 – 537.

**Reference:**

Matsui, H., Y. Kondo, N. Moteki, N. Takegawa, L. K. Sahu, Y. Zhao, H. E. Fuelberg, W. R. Sessions, G. Diskin, D. R. Blake, A. Wisthaler, and M. Koike: Seasonal variation of the transport of black carbon aerosol from the Asian continent to the Arctic during the ARCTAS aircraft campaign, J. Geophys. Res., 116, D05202, doi:10.1029/2010JD015067, 2011.

**Response to Reviewer #2's Comments**

Anonymous Referee #2:

This study investigated the responses of Arctic BC concentrations and surface temperature to 20% anthropogenic emission reductions from six source regions (East Asia, Europe, Middle East, North America, Russia-Belarus-Ukraine, and South Asia) by using several global aerosol models. The contribution from East Asia was estimated to be the largest (18.1-54.1%) among the six source regions. The authors also showed that the source contributions of the Arctic BC vary with altitude and latitude and the responses of the Arctic BC to emission reductions from the six source regions decrease with latitude. Finally, this study indicated that the reductions of global BC emissions are important for the Arctic climate by showing that the response of the Arctic air temperature to BC reduction is larger than that of the global air temperature.

The results of this study are useful for understanding the current status of the estimation of the source contributions of Arctic BC by global aerosol models. However, as shown below, there are some important problems that need to be clearly described.

We sincerely thank for the reviewer's in-depth comments and helpful suggestions on this manuscript. Based on the specific comments, we have responded to all the comments point-by-point and made corresponding changes in the manuscript as highlighted in red color. The reviewer has raised a number of issues and we quite agree. We feel the substantial revisions based on the reviewer's comments have greatly improved the quality of this manuscript. Please check the detailed responses to all the comments as below.

Major comments:

1)

This study uses six global models. Two models (CHASER) overestimate the BC concentrations in the Arctic by a factor of two, and four models (CAMchem, GEOS-chem, GOCART, and OsloCTM3) underestimate BC by more than 50%. The authors use the averages of these six model results and describe that the averages reproduce the observed BC concentrations in the Arctic well (lines 268-271). However, none of these models can simulate Arctic BC well. It is not meaningful to show as if the agreement with observations was improved by averaging the multiple model results that cannot reproduce the Arctic BC. The authors need to describe more clearly why it makes sense to use the averages of the model results that cannot reproduce the observations well.

Response: Thank for the comment. We do agree with the reviewer that currently no single model could reproduce the BC concentrations over different regions of the Arctic well. There is a number of reasons responsible for this. First, the BC emission inventory in the Arctic is not well understood due to lacking of local activity data and emission factors, e.g. gas flaring in the oil and gas production fields, biofuel combustion, non-road transportation, etc. Secondly, the lifetime of BC in the atmosphere is sensitive to its wet deposition rates. However, different models have divergent treatment of wet scavenging parameterizations (Bourgeois et al., 2011; Liu et al., 2011), which may be not representative in the Arctic region and could result in the simulated BC concentrations ranging between several magnitudes. The mechanism of BC sinks is still not well understood in the Arctic. Last but not the least, almost all the global models used the latitude/longitude projection which has very large distortions over the polar regions and this may also affect the ability of global models simulating the air pollutants over the Arctic region.

In a previous study by Shindell et al. (2008), a similar ensemble modeling study on Arctic BC was conducted. As shown in the figure below, the single model cannot reproduce the observed BC monthly variations at two sites, either. As a comparison, this study showed better model performances as seen in Figure S2a, which was due to a better global BC emission inventory and development of some key physical schemes in some global models after years. However, some similar issues as previous studies still existed, such as overestimation of BC during summer and underestimation of BC during winter. To reduce the bias from one single model, the best way may be using the ensemble model mean as similar as those climate studies such as CMIP5 and CMIP6. This is also the goal of HTAP that collects various global model simulation results of atmospheric chemistry and uses the model ensemble results to solve the source-receptor relationship in regions of interest.

| Model | Gas-phase | Aerosols | Prescribed lifetime | Horizontal Resolution |
|---|---|---|---|---|
| 1. CAMCHEM | $NO_x$, CO | SO2, BC | Y | 1.9 |
| 2. ECHAM5-HAMMOZ | | SO2, BC | | 2.8 |
| 3. EMEP | $NO_x$, CO | SO2 | | 1.0 |
| 4. FRSGC/UCI | $NO_x$, CO | | Y | 2.8 |
| 5. GEOSChem | $NO_x$ | SO2, BC | | 2.0 |
| 6. GISS-PUCCINI | $NO_x$, CO | SO2, BC | Y | 4.0 |
| 7. GMI | $NO_x$, CO | SO2, BC | Y | 2.0 |
| 8. GOCART-2 | | SO2, BC | | 2.0 |
| 9. LMDz4-INCA | | SO2, BC | | 2.5 |
| 10. LLNL-IMPACT | $NO_x$, CO | SO2, BC | | 2.0 |
| 11. MOZARTGFDL | $NO_x$, CO | SO2, BC | Y | 1.9 |
| 12. MOZECH | $NO_x$, CO | | Y | 2.8 |
| 13. SPRINTARS | | SO2, BC | | 1.1 |
| 14. STOCHEM-HadGEM1 | $NO_x$, CO | | | 3.8 |
| 15. STOCHEM-HadAM3 | $NO_x$, CO | SO2 | Y | 5.0 |
| 16. TM5-JRC | $NO_x$ | SO2, BC | | 1.0 |
| 17. UM-CAM | $NO_x$, CO | | Y | 2.5 |

2)

As described in the comment (1), two models (CHASER) overestimate Arctic BC by a factor of 2 and four models underestimate it by more than 50%. Therefore, the two CHASERs have about 4 times higher concentrations than the other four models. Then, the simple averages of these models strongly depend on the results of the two CHASERs with their contributions of about 2/3 to the total (CAMchem: CHASER_re1: CHASER_t106: CEOS-chem: GOCART: OsloCTM3 = 1: 4: 4: 1: 1: 1). If this is correct, the multi-model averages shown in this study will not be very different from the results of CHASER (the contributions of the other models will be too small).

Considering these points, I suggest the authors to revise the manuscript for the three points below. First, please discuss the spatial distributions and source contributions of BC in the Arctic for each model. Second, please use one CHASER, not two. Third, please consider using medians instead of averages, as has been done in other multi-model studies.

Response: Thanks for the suggestion. We have made the following changes as below.

(1) As suggested, we have deleted the simulation results of CHASER_t106 and kept the results of CHASER_re1. Since there were only five models used in this study, the medians may represent the results from one model. As no single model can well reproduce the observations, we think it should be more practicable by using the mean values of the five models. In the revised manuscript, we have changed all the calculations based on results from the five models.

(2) We have added more discussions on the spatial distributions and source contributions of BC in the Arctic for each model as below.

Spatial distributions of Arctic near surface BC concentrations in summer and winter simulated from each model are showed in Figure S4. BC simulated by CHASER_re1 showed relatively high concentrations over the whole Arctic, followed by GEOS-chem and GOCART–v5, while those simulated by Oslo CTM3–v2 and CAMchem were lower. The difference of simulated BC concentrations between land and ocean was more obvious in summer than that in winter, especially for GEOS-chem and GOCART–v5. The coverage of the polar dome expanded more southward in winter (Bozem et al.,

2019; Law and Stohl, 2007), allowing more BC from lower latitudinal regions to be transported into the Arctic. Turbulent exchange and deposition were reduced during winter as the meteorological conditions in the Arctic were stable and dry (Bradley et al., 1992; Bozem et al., 2019; Law and Stohl, 2007). In addition, BC emissions in EAS, EUR, and RBU regions showed obvious monthly changes with higher emissions from November to March as mentioned earlier (Section 2.1.1), leading to the relatively high BC concentrations over the polar sea ice region in winter. All models showed that the simulated BC concentrations in summer were higher than those in winter in the Russian Far East and U.S. Alaska regions due to intense biomass emissions in June, July, and August (Fig. S1).

[Figure]

(a) CAMchem – summer      (b) CAMchem – winter

(c) CHASER_re1 – summer      (d) CHASER_re1 – winter

(e) GEOS-chem – summer  (f) GEOS-chem – winter

(g) GOCART-v5 – summer  (h) GOCART-v5 – winter

(i) Oslo CTM3-v2 – summer  (j) Oslo CTM3-v2 – winter

**Figure S4.** Spatial distributions of Arctic near surface BC concentrations in summer and winter simulated from each model.

The contributions of 20% emission reductions to Arctic near surface BC concentrations simulated by different models show similar monthly variations (Figure S6). Among the five models, CHASER_re1 simulated high BC concentrations

compared to the other models due to slow aging-speed (Sudo et al., 2015). All models showed the major source regions of Arctic BC from EAS, EUR, and RUB. NAM and SAS contributed moderately while the contribution from MDE was negligible.

[Figure]

**Figure S6.** Monthly reduced concentrations of the Arctic near-surface BC due to 20% emission reductions from six source regions for each model in 2010.

We have added the discussions on the comparison among different models in line 301 – 309 and 386 – 391.

3)

Section 3.3.2 discusses the vertical profiles of Arctic BC. However, they are not validated by observations. Because several aircraft measurements such as HIPPO, ATom, and ARCTAS are available, the authors should use such data to validate the simulated vertical profiles of Arctic BC.

Response: Thanks for the suggestion. The vertical profiles of simulated BC concentrations of the BASE simulation were also compared with aircraft measurements from HIAPER Pole-to-Pole Observations (HIPPO) during 24 March–16 April 2010 (Figure S2b). Different from comparison between observed and simulated BC concentrations near the surface, the vertical profiles of BC concentrations were overestimated by most models. As the aircraft ascended and descended along each flight track, BC concentrations from HIPPO varied with time, latitude, longitude, and altitude. However, most of the simulation results of HTAP2 were provided in the temporal resolution of monthly, simulation and observation results cannot be exactly matched. This partly explained the difference between the simulations and observations. In general, the model ensemble mean could capture the vertical pattern of observed BC profiles.

[Figure]

(b)

**Figure S2 (b)** Comparison between the vertical profiles of simulated (BASE simulation) and observed BC concentrations (HIPPO) during 24 March–16 April 2010.

We have added the discussion on the model evaluation against aircraft observations in line 278 – 287.

4)

Several observational and trajectory-based studies have shown that the contribution from Asia (especially South Asia) to the near-surface BC concentration in the Arctic is limited (e.g., Stohl, 2006; Matsui et al., 2011). Please clarify whether the results of this study are consistent with the findings of these previous studies.

Stohl (2006): doi:10.1029/2005JD006888, Matsui (2011): doi:10.1029/2010JD015067

Response: Thanks for the suggestion. The comparisons between our results and previous studies are presented below.

(1) Comparison with Stohl (2006)

In Stohl (2006), the transport pathways of pollution from Europe to the low levels of the Arctic were more than that from Asia (especially South Asia). Table R1 in Stohl (2006) showed that although the contributions from South Asia to BC concentrations north of 80°N were smaller than that from Europe in low altitudes (0–5km), the contributions from South Asia were larger than that from Europe in high altitudes (5–12km). This was consistent with our conclusion (take Figure 5 for example).

**Table R1.** Average contributions (ng m$^{-3}$) from Europe and South Asia to BC concentrations north of 80°N after 30 days of transport, for Januaries and Februaries in the years 2000-2005.

| Layer | Europe | South Asia |
|---|---|---|
| 0−0.1 km | 102 | 19 |
| 0.1−0.5 km | 106 | 22 |
| 0.5−1 km | 107 | 27 |
| 1−2 km | 92 | 34 |
| 2−3 km | 69 | 40 |
| 3−5 km | 42 | 43 |
| 5−7 km | 18 | 40 |
| 7−10 km | 4 | 23 |
| 10−12 km | 0.2 | 2.7 |

(2) Comparison with Matsui (2011)

Matsui et al. (2011) mainly compared the influence of biomass burning (BB) in Russia (Russian BB) and anthropogenic emissions (AN) in East Asia (Asian AN) on BC over the North American Arctic. Our paper aims to investigate the responses of Arctic BC concentrations to anthropogenic emissions from different regions. There are two differences between Matsui et al. (2011) and this study. One is the difference of emission sources. Matsui et al. (2011) evaluated impacts from both biomass burning emissions and anthropogenic emissions, while the HTAP experiment only focused on sensitivity simulations of anthropogenic emissions. And the other is the difference of receptors. Matsui et al. (2011) quantified the contributions of emissions to BC over the North American Arctic, while we investigated the response of BC over the whole Arctic to emissions. Since fuel types of combustion and receptor regions are different, it is hard to compare the results between Matsui et al. (2011) and our study.

Matsui (2011) pointed out that Asian AN air masses were measured most frequently in the upper troposphere, with median values of 20 ng m$^{-3}$ (410hPa) in April 2008 and 5 ng m$^{-3}$ (353hPa) in June–July 2008. In our analysis, the contribution of 20% emission from EAS and SAS to BC in the Arctic was 1.4 ng m$^{-3}$ (432hPa) in April 2010 and 0.7 ng m$^{-3}$ (375hPa) in June–July 2010. If the contribution is linearly interpolated, the contribution of 100% emission from EAS and SAS to BC in the Arctic would be about 7 ng m$^{-3}$ (432hPa) in April and 3.5 ng m$^{-3}$ (375hPa) in June–July in 2020. In general, our results were at the same magnitude with Matsui (2011).

In the revision, we have added the comparison between this study and previous studies.

Specific comments:

5) Lines 35 and 37

18.1 – 54.1%: Please clarify the meaning of the range.

Response: The percentages (18.1%–51.4%) mean the ratios between the response of Arctic near-surface BC concentrations to emission reductions from EAS versus that from all six regions. To avoid confusion, we changed percentages to mass concentrations.

In the revision, we have changed the original sentence as "Emission reductions from East Asia (EAS) had most (monthly contributions: 0.2 - 1.5 ng m$^{-3}$) significant impact

on the Arctic near surface BC concentrations while the monthly contributions from Europe (EUR), Middle East (MDE), North America (NAM), Russia-Belarus-Ukraine (RBU), and South Asia (SAS) were 0.2–1.0 ng m$^{-3}$, 0.001–0.01 ng m$^{-3}$, 0.1–0.3 ng m$^{-3}$, 0.1–0.7 ng m$^{-3}$, 0.0–0.2 ng m$^{-3}$, respectively."

6) Lines 118 – 119

Please add the amount of global BC emission flux from anthropogenic sources and describe how different it is from the CMIP6 emissions.

Response: Thanks for the suggestion. The amount of global BC anthropogenic emissions was 5.5 Tg/year in 2010 from the HTAP2 emissions. The amount of global BC anthropogenic emissions was 7.7 Tg/year in 2010 from the CMIP6 emissions (Hoesly et al., 2018). Compared with CMIP6 emissions, global BC anthropogenic emissions of HTAP2 was about 30% lower. This was mainly due to the energy, transportation, and international shipping sectors of CMIP6 were higher than those of HTAP2. We have added these values. Please see the discussions in lines 552 – 557 in the revision.

7) Lines 141 – 144, Table 1

Please describe how the authors determined the monthly variations of BC emissions. Please add the values of global emissions.

Response: HTAP2 established a global-scale BC emission inventory by compiling several regional gridded inventories. Data sources included BC emissions of the Environmental Protection Agency (EPA) for USA, the EPA and Environment Canada (for Canada), the European Monitoring and Evaluation Programme (EMEP) and Netherlands Organisation for Applied Scientific Research (TNO) for Europe, the Model Inter-comparison Study for Asia (MICS-Asia III) for China, India and other Asian countries, and the Emissions Database for Global Atmospheric Research (EDGARv4.3) for the rest of the world (mainly South America, Africa, Russia and Oceania). Temporal resolution of data sources was monthly, and thus the HTAP2 emission inventory provided harmonized emission data with monthly resolution for all the air pollutants including BC. It should be noted that the emissions of international shipping and international aviation in HTAP2 were considered constant over the year. Monthly BC data can be download from https://edgar.jrc.ec.europa.eu/htap_v2/index.php?SECURE=123.

The values of global emissions have been added in Table 1. Please check the updated Table 1 in the revision.

8) Line 209 What is "m" in equation (1).

Response: "m" is the latitude band which the receptor region (refers to the Arctic in this paper) is located. Its meaning has already been defined in line 211 – 213.

9) Line 314

1.0 – 1.7 ng m$^{-3}$, 1.9 – 3.8 ng m$^{-3}$: Please clarify the meaning of the ranges.

Response: The ranges of the concentrations mean the monthly contribution of BC emission reductions from all six source regions (the sum of EAS, EUR, MDE, NAM, RBU, and SAS) to Arctic near-surface BC concentrations. We have stated more clearly in lines 341 – 343.

10) Line 329

The sum of the six sources (from 20% emission reductions) is about 2 ng m$^{-3}$. I suspect that the contributions from sources other than the six regions (e.g., anthropogenic sources in the Arctic, biomass burning sources) are large.

Response: Thanks for the comment. 2 ng m$^{-3}$ was the contribution of 20% BC emission reductions from the six source regions to the Arctic near-surface BC concentrations. By assuming that BC is an inert particulate component, the contribution of 100% BC emissions from six regions to the Arctic near-surface BC concentrations was about five times of this value (close to 10 ng m$^{-3}$). The annual mean Arctic near-surface BC concentration from the BASE simulation was about 18 ng m$^{-3}$ in 2010. By comparing the values of contribution from all six regions (10 ng m$^{-3}$) and BC concentration in the Arctic (18 ng m$^{-3}$), the impact of emissions from six regions on the Arctic near-surface BC was outstanding. It should be noted that the contributions from six regions only considered anthropogenic emissions while the contribution from biomass burning was not included in the sensitivity experiments of HTAP2. It is known that wildfires in Fast East of Russia and U.S. Alaska are important sources of BC in the Arctic region, especially in summer. Thus, the contributions from six regions to the Arctic BC should be even more dominate over the other regions by including biomass burning in RBU and NAM.

11) Line 333

0.3 – 1.9 ng m-3, 18.1 – 51.4%: Please clarify the meaning of the ranges. Please show the contributions of anthropogenic and biomass burning sources other than the six sources. Are the contributions of the six sources to total BC in the Arctic dominant?

Response: 0.3 – 1.9 ng m$^{-3}$ means that "Arctic near-surface monthly BC concentrations can be decreased by 0.3–1.9 ng m$^{-3}$, resulting from the 20% anthropogenic emission reduction from EAS. The percentages (18.1%–51.4%) mean the ratios between the response of Arctic near-surface BC concentrations to emission reductions from EAS

versus that from all six regions. We have changed this sentence as "The response of Arctic near-surface monthly BC concentration was found strongest to the 20% emission reductions from EAS with the contribution of 0.2–1.5 ng m$^{-3}$, accounting for 16.8%– 49.0% of the total reduced BC concentrations resulting from all six source regions".

As stated in the last response, HTAP2 focused only on anthropogenic emissions by conducting sensitivity experiments from different source regions based on various models. However, no sensitivity experiments on biomass burning emissions were designed. Hence, we cannot show the contributions of anthropogenic and biomass burning sources other than the six sources in this study.

As also stated in the last response, although HTAP only set experiments by reducing 20% of the anthropogenic emissions, we can estimate that the contributions of the six sources to total BC in the Arctic were more than a half if 100% emissions from all six source regions were cut off. In addition, by including the biomass burning emissions in RBU and NAM, the contributions from those six regions would be even greater. In this regard, we believe the six sources are important source regions to the Arctic. And we do agree with the reviewer that the contributions from some other regions such as Africa should be also evaluated in the future study.

12) Lines 342 – 343

Please clarify the meaning of the ranges.

Response: We have made this sentence more clearly in lines 370 – 373 as "In addition to EAS, BC emission reduction from EUR also showed significant impacts on the Arctic near-surface monthly BC concentration with the contribution of 0.2–1.0 ng m$^{-3}$, accounting for 20.1%–49.0% of the total reduced BC concentrations resulting from all six source regions."

13) Line 356

Section 3.3.2: the vertical profiles of BC should be validated by the available aircraft measurements.

Response: As responded in Q3, the vertical profiles of simulated BC concentrations of the BASE simulation were compared with aircraft measurements from HIPPO. In general, the model ensemble mean results can reproduce the observed profiles. We have added this validation result in line 278– 287.

14) Line 433

How important are these two peaks? Is this consistent with observations?

Response: Thanks for the comment. The two peaks reflected the dominance of different source regions to the BC profiles at different altitudes. As stated previously, all the simulation results in this study were based on 20% emission reductions but not 100% emission reductions. While the observations were related to all the anthropogenic and biomass burning emissions, it is hard to compare the results from sensitivity simulations and observations. Also, the aircraft observations were episodic while the simulation results were presented in the seasonal scale. It is also hard to match the simulation and observation.

15) Lines 449 – 451

Please clarify how important these concentrations are.

Please show the contributions of these sources to total BC.

Response: This paragraph analyzed the differences of contributions of emission reductions to BC in eight latitudinal bands during summer and winter, which focused on the seasonal variation. These concentrations clarified the benefits of ambient BC reductions in different latitudinal bands of the Arctic contributed from the 20% emission reductions.

16) Line 457

10 years: What does this mean? The time scale of BC impacts on Arctic temperatures is probably much shorter than 10 years.

Response: BC plays an important role in climate system through absorption of solar radiation, interaction with clouds, and deposition on snow and ice. We quite agree with the reviewer that the lifetime of BC in the atmosphere is much shorter than 10 years. While in the Arctic region which is mostly covered by snow and ice, the deposition of BC on the snow and ice still affects the Arctic temperature by changing the surface albedo. In the calculation of ARTP, the enhanced climate efficacy for BC deposition on snow has been taken into account. Aamaas et al. (2017) also showed that the ARTP20 of BC deposition on snow was larger than that of other effects in winter (Figure R4 b).

[Figure]

**Figure R2.** ARTP20 for BC emissions from Europe, East Asia, shipping, and global emissions for summer and winter. In each frame and for each emission region, the ARTP20 values for the four latitudinal response bands from south (left) to north (right), as well as the global response average (rightmost) for the species, are decomposed by effects. The net response is shown by the asterisk. The regions included are Europe (EUR), East Asia (EAS), shipping (SHP), and the globe (GLB), all for both NH summer, May–October (left), and NH winter, November–April (right).

17) Lines 460 – 461

What do these temperatures mean? It's unclear how they correspond to actual temperature changes.

Response: The temperatures here means Arctic surface temperature responses to emission reductions from source regions. It can be calculated by the following formula:

$$\Delta T_{r, m, s, t}(t) = \int_0^t E_{r, s, t}(t') \times ARTP_{r,m,s, t}(t-t')dt'$$

$\Delta T_{r,m,s,t}$ refers to the decrease of the Arctic or global surface temperature after 20, 100, or 500 years to 20% BC emission reductions of six regions (namely EAS, EUR, MDE, NAM, RBU, and SAS) in the framework of HTAP2 either during summer or winter. $E_{r,s,t}$ refers to the 20% BC emission reductions of six regions. $ARTP_{r,m,s,t}$ refers to Absolute Regional Temperature-change Potential. $r$ refers to source region, $m$ refers to the latitude band of receptor region (Arctic here), $s$ refers to seasons, $t$ refers to time horizons.

The scenario of HTAP2 was 20% emission reduction from all anthropogenic emission sectors. However, in reality, not all emissions sectors of a specific source region cannot be reduced by 20% at the same time. In other words, responses of Arctic surface temperature to 20% emission reductions are more suitable to be used for the comparison

among different source regions but cannot be used to reflect the actual change of temperature.

In addition, there are many other factors (e.g. greenhouse gases, sea ice coverage) that can affect the temperature change in the Arctic besides BC. BC may be one of the factors affecting the ambient temperature but probably not the dominant one. Thus, we didn't compare the temperature change caused by BC emission reductions from six source regions with actual temperature change.

18) Line 475

The authors conclude that South Asia will be important. Is this consistent with previous studies (e.g., Stohl, 2006)?

Response: Stohl (2006) showed that BC contributed from South Asia were larger than that from Europe at high altitudes (5–12km). Our results were consistent with Stohl (2006). In Line 475, we meant to roughly predict the role of South Asia in the future Arctic. Compared to East Asia that is experiencing obviously decreasing air pollutants emission, South Asia is now becoming a blooming economy. A number of studies have shown that BC emissions in SAS increased in recent years (Sahuet al., 2008; Paliwal et al., 2016; Sharma et al., 2019). The role of South Asia in regulating the Arctic climate is expected to be more and more important.

19) Line 522

Are the two CHASERs different enough to be considered as different models?

Response: As the reviewer suggested, we have deleted the simulation results of CHASER_t106 and re-analyzed all the results.

20) Line 534

Is the correlation coefficient of 0.98 due to the two CHASERs? I think the value is too high as a correlation between independent models.

Response: We have double checked that the correlation coefficient of 0.98 was obtained by comparing the monthly BC concentrations at Alert simulated by GEOS-chem and GOCART–v5.

21) Lines 536 – 537

I don't think it makes sense to show as if the agreement with observations was improved by averaging the multiple models that overestimate and underestimate observations. None of the models in this study can simulate Arctic BC reasonably well.

Response: Thanks for the comment. We quite agree with the reviewer. As expressed in response of Q1 and Q2 in major comments, averaging the multiple models was the most desirable method at present. Of course, the simulation of the model is constantly improving, and there may be more suitable simulation schemes or even models for the Arctic in the future. We have changed our expression. Please see the conclusions in lines 591 – 592 in the revision.

22) Line 543

Please clarify the meaning of the range.

Response: We have stated it more clearly as "The response of Arctic near-surface BC concentrations to 20% emission reductions from EAS and EUR was larger than other four source regions, with the monthly value of 0.2–1.5 ng m$^{-3}$ and 0.2–1.0 ng m$^{-3}$, accounting for 16.8%–49.0% and 20.1%–49.0% of the total contributions from all six regions, respectively." in lines 597 – 600.

23) Lines 545 – 547

Is this consistent with observations?

Response: Thanks for the comment. As responded in Q14, the simulation results represented the contributions from 20% anthropogenic emission reduction while the observational results were related to all the emissions. Thus, it was not feasible to compare the simulated vertical BC contributions with observations.

**Reference:**

Aamaas, B., Berntsen, T. K., Fuglestvedt, J. S., Shine, K. P., and Collins, W. J.: Regional temperature change potentials for short-lived climate forcers based on radiative forcing from multiple models, Atmos. Chem. Phys., 17, 10795-10809, 10.5194/acp-17-10795-2017, 2017.

Bond, T. C., D. G. Streets, K. F. Yarber, S. M. Nelson, J.-H. Woo, and Z. Klimont: A technology-based global inventory of black and organic carbon emissions from combustion, J. Geophys. Res., 109, D14203, doi:10.1029/2003JD003697, 2004.

Bozem, H., Hoor, P., Kunkel, D., Köllner, F., Schneider, J., Herber, A., Schulz, H., Leaitch, W. R., Aliabadi, A. A., Willis, M. D., Burkart, J., and Abbatt, J. P. D.: Characterization of transport regimes and the polar dome during Arctic spring and summer using in situ aircraft measurements, Atmos. Chem. Phys., 19, 15049-15071, 10.5194/acp-19-15049-2019, 2019.

Chen, D. S., Zhao, Y. H., Nelson, P., Li, Y., Wang, X. T., Zhou, Y., Lang, J. L., and Guo, X. R.: Estimating ship emissions based on AIS data for port of Tianjin, China. Atmos. Environ. 145: 10–18. http://dx.doi.org/10.1016/j.atmosenv.2016.

Law, K. S. and Stohl, A.: Arctic air pollution: origins and impacts, Science, 315, 1537–1540, https://doi.org/10.1126/science.1137695, 2007.

Matsui, H., Y. Kondo, N. Moteki, N. Takegawa, L. K. Sahu, Y. Zhao, H. E. Fuelberg, W. R. Sessions, G. Diskin, D. R. Blake, A. Wisthaler, and M. Koike: Seasonal variation of the transport of black carbon aerosol from the Asian continent to the Arctic during the ARCTAS aircraft campaign, J. Geophys. Res., 116, D05202, doi:10.1029/2010JD015067, 2011.

Paliwal, U., Sharma, M., and Burkhart, J. F.: Monthly and spatially resolved black carbon emission inventory of India: uncertainty analysis, Atmos. Chem. Phys., 16, 12457-12476, 10.5194/acp-16-12457-2016, 2016.

Sahu, S. K., Beig, G., and Sharma, C.: Decadal growth of black carbon emissions in India, Geophysical Research Letters, 35, 10.1029/2007gl032333, 2008.

Sharma, G., Sinha, B., Pallavi, Hakkim, H., Chandra, B. P., Kumar, A., and Sinha, V.: Gridded Emissions of CO, NO x, SO2, CO2, NH3, HCl, CH4, PM2.5, PM10, BC, and NMVOC from Open Municipal Waste Burning in India, Environ Sci Technol, 53, 4765-4774, 10.1021/acs.est.8b07076, 2019.

Stohl, A.: Characteristics of atmospheric transport into the Arctic troposphere, J. Geophys. Res., 111, D11306, doi:10.1029/2005JD006888, 2006.

Sudo, K., Sekiya, T., Nagashima, T.: CHASER/MIROC-ESM in HTAP2 status reports, HTAP2 Global and Regional Model Evaluation Workshop, Nagoya University, JAMSTEC, NIES, 2015.

Hoesly R. M., Smith S. J., Feng L. Y., Klimont Z., Janssens-Maenhout G., Pitkanen T., Seibert J. J., Vu L., Andres R. J., Bolt R. M., Bond T. C., Dawidowski L., Kholod N., Kurokawa J., Li M., Liu L., Lu Z. F., Moura M. C. P., O'Rourke P. R., Zhang Q.: Historical (1750–2014) anthropogenic emissions of reactive gases and aerosols from the Community Emissions Data System (CEDS), Geosci. Model Dev., 11, 369–408, 10.5194/gmd-11-369-2018, 2018.

---

## Author Response (AR2)

**Response to Reviewer #2's Comments**

The manuscript was improved substantially by this revision. I have a few suggestions. We sincerely thank for the reviewer's time and helpful suggestions on greatly improving the quality of this manuscript. As for the further comments, we have responded to all the comments point-by-point and made corresponding changes in the manuscript as highlighted in red color. Please check the responses to all the comments as below.

1) Some discussion in the response letter should be added to the manuscript.

First, in the response to my previous review comment (1), the authors described as follows: "We do agree with the reviewer that currently no single model could reproduce the BC concentrations over different regions of the Arctic well. There is a number of reasons responsible for this. First, the BC emission inventory in the Arctic is not well understood due to lacking of local activity data and emission factors, e.g. gas flaring in the oil and gas production fields, biofuel combustion, non-road transportation, etc. Secondly, the lifetime of BC in the atmosphere is sensitive to its wet deposition rates. However, different models have divergent treatment of wet scavenging parameterizations (Bourgeois et al., 2011; Liu et al., 2011), which may be not representative in the Arctic region and could result in the simulated BC concentrations ranging between several magnitudes. The mechanism of BC sinks is still not well understood in the Arctic. Last but not the least, almost all the global models used the latitude/longitude projection which has very large distortions over the polar regions and this may also affect the ability of global models simulating the air pollutants over the Arctic region. In a previous study by Shindell et al. (2008), a similar ensemble modeling study on Arctic BC was conducted. As shown in the figure below, the single model cannot reproduce the observed BC monthly variations at two sites, either. As a comparison, this study showed better model performances as seen in Figure S2a, which was due to a better global BC emission inventory and development of some key physical schemes in some global models after years. However, some similar issues as previous studies still existed, such as overestimation of BC during summer and underestimation of BC during winter. To reduce the bias from one single model, the

best way may be using the ensemble model mean as similar as those climate studies such as CMIP5 and CMIP6. This is also the goal of HTAP that collects various global model simulation results of atmospheric chemistry and uses the model ensemble results to solve the source-receptor relationship in regions of interest.". This discussion can be added to the manuscript.

Thanks for the suggestion. We have added a paragraph in Line 294 – 304.

Second, in the response to my previous review comment (4), the authors described as follows: "Matsui (2011) pointed out that Asian AN air masses were measured most frequently in the upper troposphere, with median values of 20 ng m$^{-3}$ (410hPa) in April 2008 and 5 ng m$^{-3}$ (353hPa) in June–July 2008. In our analysis, the contribution of 20% emission from EAS and SAS to BC in the Arctic was 1.4 ng m$^{-3}$ (432hPa) in April 2010 and 0.7 ng m$^{-3}$ (375hPa) in June–July 2010. If the contribution is linearly interpolated, the contribution of 100% emission from EAS and SAS to BC in the Arctic would be about 7 ng m$^{-3}$ (432hPa) in April and 3.5 ng m$^{-3}$ (375hPa) in June–July in 2020. In general, our results were at the same magnitude with Matsui (2011)." The authors described that the comparison between the author's study and previous studies has been added to the manuscript, but I could not find this discussion in the revised manuscript.

Thanks for the suggestion. We have added a paragraph in Line 452 – 459.

Third, in the response to my previous review comment (7), the authors described as follows: "Temporal resolution of data sources was monthly, and thus the HTAP2 emission inventory provided harmonized emission data with monthly resolution for all the air pollutants including BC. It should be noted that the emissions of international shipping and international aviation in HTAP2 were considered constant over the year." This information can be added to the manuscript.

Thanks for the suggestion. We have added it in Line 123 – 126.

Fourth, in the response to my previous review comment (10), the authors described as follows: "2 ng m$^{-3}$ was the contribution of 20% BC emission reductions from the six

source regions to the Arctic near-surface BC concentrations. By assuming that BC is an inert particulate component, the contribution of 100% BC emissions from six regions to the Arctic near-surface BC concentrations was about five times of this value (close to 10 ng m$^{-3}$). The annual mean Arctic near-surface BC concentration from the BASE simulation was about 18 ng m$^{-3}$ in 2010. By comparing the values of contribution from all six regions (10 ng m$^{-3}$) and BC concentration in the Arctic (18 ng m$^{-3}$), the impact of emissions from six regions on the Arctic near-surface BC was outstanding. It should be noted that the contributions from six regions only considered anthropogenic emissions while the contribution from biomass burning was not included in the sensitivity experiments of HTAP2. It is known that wildfires in Fast East of Russia and U.S. Alaska are important sources of BC in the Arctic region, especially in summer. Thus, the contributions from six regions to the Arctic BC should be even more dominate over the other regions by including biomass burning in RBU and NAM." I suggest the authors add these results to the manuscript.

Thanks for the suggestion. We have added a paragraph in Line 377 – 386.

Fifth, in the response to my previous review comment (17), the authors described as follows: "The scenario of HTAP2 was 20% emission reduction from all anthropogenic emission sectors. However, in reality, not all emissions sectors of a specific source region cannot be reduced by 20% at the same time. In other words, responses of Arctic surface temperature to 20% emission reductions are more suitable to be used for the comparison among different source regions but cannot be used to reflect the actual change of temperature. In addition, there are many other factors (e.g. greenhouse gases, sea ice coverage) that can affect the temperature change in the Arctic besides BC. BC may be one of the factors affecting the ambient temperature but probably not the dominant one. Thus, we didn't compare the temperature change caused by BC emission reductions from six source regions with actual temperature change.". These sentences should be added to the manuscript. The authors should clarify more why the analysis using ARTP is meaningful and where readers should focus on in this analysis.

Thanks for the suggestion. We have added a paragraph as below in Line 209 – 214.

The ARTP provide additional insight into the spatial pattern of temperature response to

inhomogeneous forcings beyond that available from traditional global metrics. Very few metrics have attempted to examine sub-global scales thus far, though some have used local information with non-linear global damage metrics (Shine et al., 2005a; Lund et al., 2012). Shindell et al. (2012) indicated that the forcing/response portion of the ARTP appeared to be relatively robust across models.

2) Please clarify why the statistics in this study are shown with ranges. What do these ranges mean? Model variability, spatial variability in the Arctic, or monthly variability? Response: The ranges were calculated based on different statistical categories.

a) Model variability:

Lines 275 – 277: Relatively good agreement between the observation and models was found at Zeppelin, with CORs, NME, MB, and MAE of 0.59–0.83, 38.59%–142.64%, –13.53–14.97 ng m$^{-3}$, and 5.40–14.97 ng m$^{-3}$, respectively

This sentence is changed as "Relatively good agreement between the observation and models was found at Zeppelin, with CORs, NME, MB, and MAE of 0.59–0.83, 38.59%–142.64%, –13.53–14.97 ng m$^{-3}$, and 5.40–14.97 ng m$^{-3}$ among the five models, respectively"

b) Monthly variability:

Lines 34 – 39: Emission reductions from East Asia (EAS) had most (monthly contributions: 0.2–1.5 ng m$^{-3}$) significant impact on the Arctic near surface BC concentrations while the monthly contributions from Europe (EUR), Middle East (MDE), North America (NAM), Russia-Belarus-Ukraine (RBU), and South Asia (SAS) were 0.2–1.0 ng m$^{-3}$, 0.001–0.01 ng m$^{-3}$, 0.1–0.3 ng m$^{-3}$, 0.1–0.7 ng m$^{-3}$, 0.0–0.2 ng m$^{-3}$, respectively.

Lines 359 – 361: The contributions of 20% BC emission reductions from all six regions to Arctic near-surface BC concentrations were 0.8–1.4 ng m$^{-3}$ during May to October and 1.5–3.2 ng m$^{-3}$ during November to April.

We have added "monthly" ahead of "Arctic near-surface BC concentrations" for clarification.

Lines 386 – 389: The response of Arctic near-surface monthly BC concentration was

found strongest to the 20% emission reductions from EAS with the contribution of 0.2–1.5 ng m$^{-3}$, accounting for 16.8%–49.0% of the total reduced BC concentrations resulting from all six source regions.

This sentence is changed as "The response of Arctic near-surface BC concentration was found strongest to the 20% emission reductions from EAS with the monthly contribution of 0.2–1.5 ng m$^{-3}$, accounting for 16.8%–49.0% of the total reduced BC concentrations resulting from all six source regions"

Lines 396 – 399: In addition to EAS, BC emission reduction from EUR also showed significant impacts on the Arctic near-surface monthly BC concentration with the contribution of 0.2–1.0 ng m$^{-3}$, accounting for 20.1%–49.0% of the total reduced BC concentrations resulting from all six source regions.

It is changed as "In addition to EAS, BC emission reduction from EUR also showed significant impacts on the Arctic near-surface BC concentration with the monthly contribution of 0.2–1.0 ng m$^{-3}$, accounting for 20.1%–49.0% of the total reduced BC concentrations resulting from all six source regions".

Lines 402 – 404: As for NAM and RBU, their 20% emission reductions induced moderate reductions of the monthly Arctic near-surface BC concentrations by 0.1–0.3 and 0.1–0.7 ng m$^{-3}$, respectively. It has been clearly indicated the ranges referred to monthly variations.

Lines 404 – 406: The contribution of 20% emission reductions from SAS to the Arctic near-surface BC concentrations was much lower of 0.0–0.2 ng m$^{-3}$ as a significant portion of BC originating from SAS accumulated in the upper troposphere.

It is changed as "The contribution of 20% emission reductions from SAS to the Arctic near-surface BC concentrations was much lower of monthly contributions of 0.0–0.2 ng m$^{-3}$ as a significant portion of BC originating from SAS accumulated in the upper troposphere".

Lines 637 – 640: The response of Arctic near-surface BC concentrations to 20% emission reductions from EAS and EUR was larger than other four source regions, with the monthly value of 0.2–1.5 ng m$^{-3}$ and 0.2–1.0 ng m$^{-3}$, accounting for 16.8%–49.0%

and 20.1%–49.0% of the total contributions from all six regions, respectively.

It has been clearly indicated the ranges referred to monthly variations.

c) Variation of latitude bands

Line 496 – 499: In contrast, the contributions from EAS (0.3–0.4 ng m$^{-3}$ in summer and 0.9–1.1 ng m$^{-3}$ in winter) were higher than those from EUR (0.2–0.4 ng m$^{-3}$ in summer and 0.4–0.9 ng m$^{-3}$ in winter) in the other high latitudinal bands where long-range transport played the dominant role.

It is changed as "In contrast, the latitudinal contributions from EAS (0.3–0.4 ng m$^{-3}$ in summer and 0.9–1.1 ng m$^{-3}$ in winter) were higher than those from EUR (0.2–0.4 ng m$^{-3}$ in summer and 0.4–0.9 ng m$^{-3}$ in winter) in the other high latitudinal bands where long-range transport played the dominant role."

Lines 521 – 523: The statistical results of SAS indicated that the contribution in vertical appeared one peak at the 15$^{th}$ layer (9.7 km a.s.l.) with a value of 0.45–0.48 ng m$^{-3}$ in summer and winter.

It is changed as "The statistical results of SAS indicated that the contribution in vertical appeared one peak at the 15th layer (9.7 km a.s.l.) with values of 0.45 and 0.48 ng m–3 in summer and winter, respectively (Figure S8).".

Lines 529 – 532: The contribution of 20% emission reductions from all six source regions to BC concentrations in eight latitude bands of the Arctic near surface was 0.7–1.9 ng m$^{-3}$ in summer and 1.8–4.1 ng m$^{-3}$ in winter, respectively (Figure 6). The high BC peak at around 0.6–1.6 km a.s.l. (3$^{rd}$ – 5$^{th}$ layers) was 1.1–2.1 ng m$^{-3}$ in summer, and 2.9–4.2 ng m$^{-3}$ in winter.

It has been clearly indicated that the ranges referred to the variations in eight latitude bands and vertical layers in the first and second sentence, respectively.

d) time horizon (10, 20, 50, and 100 years)

In Section 3.4 "Benefit of BC emission reductions on the decrease of Arctic temperature", the ranges of temperature referred to the temperature decrease after 10, 20, 50, and 100 years.